# Integrative Landscape of Dry AMD Pathogenesis, Models, and Emerging Therapeutic Strategies

**DOI:** 10.3390/ijms27010202

**Published:** 2025-12-24

**Authors:** Shiva Kumar Bhandari, Sooyeun Lee, Hye Jin Kim

**Affiliations:** College of Pharmacy, Keimyung University, 1095 Dalgubeol-daero, Dalseo-gu, Daegu 42601, Republic of Korea; 1005802@kmu.kr (S.K.B.); sylee21@kmu.ac.kr (S.L.)

**Keywords:** dry age-related macular degeneration, retinal pigment epithelium, drusen, bisretinoid, animal models, visual cycle modulation

## Abstract

Dry age-related macular degeneration (AMD) is the leading cause of central vision loss among the elderly, yet no curative treatment exists. While exudative AMD can be managed with anti-vascular endothelial growth factor (VEGF) therapy, dry AMD—accounting for more than 85% of cases—progresses insidiously from drusen accumulation to geographic atrophy (GA). Although the recent U.S. Food and Drug Administration (FDA) approvals of pegcetacoplan and avacincaptad pegol represent major milestones, their therapeutic effects remain modest. This review provides an integrated overview of the molecular and cellular mechanisms underlying dry AMD, highlighting key pathogenic pathways involving oxidative stress, lipid dysregulation, complement activation, mitochondrial impairment, and RPE-specific bisretinoid lipofuscin accumulation. We further summarize mechanistic mouse models that replicate these pathological processes and discuss how each model contributes to understanding the disease. Finally, we review current and emerging therapeutic strategies—including complement inhibitors, visual cycle modulators, and mitochondrial-protective approaches—and outline future directions for translational research. Collectively, this review synthesizes mechanistic insights, disease models, and therapeutic innovation to support the development of targeted and stage-specific interventions for dry AMD.

## 1. Introduction

Age-related macular degeneration (AMD) is a multifactorial disorder of the central retina and a leading cause of irreversible vision loss among the elderly. The disease primarily affects the retinal pigment epithelium (RPE), photoreceptors (PRs), and choriocapillaris. Early AMD is characterized by the accumulation of extracellular deposits (drusen) between the RPE and Bruch’s membrane (BrM), which may progress to either geographic atrophy (GA) in dry AMD or choroidal neovascularization in wet AMD [1].

Dry AMD represents approximately 85% of all AMD cases. Although it progresses more slowly than the neovascular form, it remains a major contributor to visual impairment and currently lacks any curative treatment [2]. Early pathological features include drusen formation and bisretinoid (RPE lipofuscin) accumulation within the RPE. In advanced stages, degeneration of both RPE and photoreceptors leads to GA, marked by irreversible loss of central vision [3,4]. With the global prevalence of AMD projected to reach 288 million individuals by 2040, a comprehensive understanding of early pathogenic events and the development of effective therapeutic strategies are urgently needed. Animal models, particularly mouse models, have played a pivotal role in elucidating AMD pathogenesis [5]. Whereas AMD involves concurrent accumulation of bisretinoids and sub-RPE deposits, many inherited maculopathies exhibit a predominance of one of these features [6,7]. Although mice lack a macula, they share approximately 90% genetic homology with humans and develop age-related retinal changes, including histological alterations and electroretinography (ERG) deficits by mid-life. Their relatively short lifespan and cost-effectiveness make them suitable for modeling progressive retinal degeneration [8]. In the context of AMD, lipofuscin refers to a heterogeneous mixture of autofluorescent pigments that accumulate over time, primarily within the lysosomal compartment of the RPE. RPE lipofuscin originates in part from the phagocytosis of shed photoreceptor outer-segment membranes, which contain vitamin A-derived adducts during the visual cycle [7]. Bisretinoids constitute a chemically defined family of vitamin A-derived fluorescent molecules formed non-enzymatically in photoreceptor outer segments through the condensation of two molecules of retinaldehyde with phosphatidylethanolamine [9,10].

Therefore, this review summarizes the pathological features of dry AMD, with emphasis on drusen and bisretinoid accumulation, and provides a critical overview of mechanistic mouse models used to interrogate these processes. Additionally, we highlight current and emerging therapeutic strategies aimed at mitigating or preventing disease progression in dry AMD.

### 1.1. Genetics Contributions to AMD

Genetic factors are estimated to account for 46–71% of AMD susceptibility. Among more than 100 associated loci, the most prominent are complement factor H (*CFH*) and age-related maculopathy susceptibility 2/high temperature requirement A serine peptidase 1 (*ARMS2/HTRA1*) [11]. Single-nucleotide polymorphisms (SNPs) in *CFH*, complement component 2 (*C2*), *C3*, and *ARMS2/HTRA1* show strong association with disease susceptibility [12,13]. The CFH Y402H variant impairs regulation of the complement cascade, resulting in excessive C3b activation and chronic inflammation [11]. Conversely, certain complement factor B (*CFB*) haplotypes, such as R32Q and R32Q/IVS10, exert protective effects against AMD [14]. Additional loci—including complement factor I (*CFI*), *HTRA1*, and Apolipoprotein E (*APOE*)—influence AMD pathogenesis through modulation of lipid metabolism, angiogenesis, and oxidative stress responses [11,15,16,17]. Table 1 summarizes major AMD-associated genes, their functional effects, phenotypic features, and translational implications.

AMD is a progressive and irreversible retinal disease that exhibits substantial heterogeneity in its clinical manifestations, progression rate, and underlying genetic susceptibilities [18]. Recognizing this variability is essential for accurate diagnosis, prognostic assessment, and the development of targeted therapeutic strategies.

Clinically, AMD is classified into early, intermediate, and late stages based on macular characteristics such as drusen size and number, as well as the presence of pigmentary abnormalities [19]. The non-neovascular (dry) form of AMD typically encompasses the early and intermediate stages of the disease and may progress through these stages prior to the development of advanced AMD. The Age-Related Eye Disease Study (AREDS) further categorizes dry AMD into four stages: Stage 1 includes few or no small drusen (<63 µm) with or without RPE changes; Stage 2 (early AMD) is defined by numerous small or few intermediate-sized drusen (63–124 µm) and/or pigment abnormalities; Stage 3 (intermediate AMD) includes many intermediate-sized or at least one large drusen (>125 µm) and/or non-central GA; Stage 4 (advanced AMD) is characterized by central GA involving the macula [20].

Genetic differences further contribute to the phenotypic diversity and progression patterns of AMD. Variants in the *CFH* gene remain among the strongest genetic determinants of AMD susceptibility [21]. Rare CFH loss-of-function variants are prominently enriched in individuals with early-onset drusen maculopathy, correlating with earlier disease onset and strong family aggregation [22,23].

Mutations in *ABCA4*, typically associated with Stargardt disease, have also been implicated in specific AMD subtypes [24]. Monoallelic *ABCA4* variants give rise to a granular AMD phenotype characterized by a fine granular peripheral pattern known as GPS (granular pattern with peripheral punctate spots) [25]. Mechanistically, ABCA4 dysfunction disrupts retinaldehyde clearance, promoting excessive bisretinoid accumulation in RPE cells and triggering oxidative stress and complement activation [26].

Similarly, variants in C3, a central regulator of the complement cascade, contribute significantly to AMD risk [27]. The common C3 rs2230199 variant is strongly associated with AMD susceptibility, while rare variants such as C3 Lys155Gin have been linked to early-onset disease and familial clustering, mirroring the risk pattern observed for CFH mutations [28].

Together, these genetic subtypes highlight distinct yet converging pathogenic pathways in dry AMD. CFH and C3 variants predominantly drive complement dysregulation, whereas ABCA4 mutations induce metabolic and oxidative stress—all ultimately contributing to RPE dysfunction and drusen formation.

**Table 1 ijms-27-00202-t001:** Key AMD-related genes: Effects, Phenotypes, Applications, and Limitations.

Gene	Effect in AMD	Human Relevance/Phenotype	Best Use	Limitation
*CFH*	Key regulator of complement system. Decrease in *CFH* level and variants increases risk of advanced AMD and drusen formation [29]	Early macular drusen [30]	To evaluate genetic risk and guide advanced AMD treatment strategies [31]	Genetic testing not advised for routine patient counseling [32]
*ARMS2*	*ARMS2* is strongly associated with AMD development, and plays important role in extracellular matrix and mitochondrial function in retinal cells [33]	*HTRA1* mutations are associated with both GA and choroidal neovascularization (CNV) [34]	Used in genetic risk assessment models for AMD prognosis.	Genetic tests often use a limited number of genes and are biased towards European ancestry [35]
*HTRA1*	*HTRA1* is strongly associated with extracellular matrix (ECM) remodeling in AMD [36]	*HTRA1* contributes to AMD, affecting drusen formation and compromising BrM [36]	Studying drusen pathogenesis and BrM function	Unable to fully replicate late-stage AMD, like GA
*C3*	Polymorphisms are associated with an increased risk of AMD, especially advanced forms. The rs2230199 (R102G) variant is considered a likely causal variant in Caucasians [37]	Involved in the complement system; specific variants can increase AMD risk	Included in predictive models for AMD prevalence and incidence [31]	The clinical importance of biomarker correlations is still unclear, requiring further study [38]
*TIMP3*	A susceptibility locus for AMD, with roles in extracellular matrix degradation	Mutations are linked to Sorsby’s fundus dystrophy, a rare form of macular degeneration	Identifying rare coding variants to pinpoint causal genes within known genetic loci [30]	Systematically identifying associations with rare variants requires extremely large sample sizes and specific study designs [30]
*APOE*	Associated with AMD and plays a role in high-density lipoprotein metabolism.	Involved in lipid transport and metabolism; the ε2 allele was the first genetic risk factor identified for AMD [39]	Included in models to assess the joint effects of genetic, ocular, and environmental variables for AMD [31]	Not all of the genetic contribution to AMD is explained by known loci, suggesting other genes with smaller effects exist [40]
*LIPC*	Influences susceptibility to AMD, associated with high-density lipoprotein cholesterol levels. Predicted higher LIPC expression in AMD cases is expected to result in lower blood HDL levels [29]	Encodes hepatic lipase (HL), which regulates HDL concentration [29]	Part of the broader genetic analysis to understand AMD etiology [29]	Single expression quantitative trait locus (eQTL) analysis has limitations, as the causative signal remains elusive for many variants [29]
*CETP*	Associated with AMD and high-density lipoprotein cholesterol levels. Lower predicted CETP expression is significantly associated with AMD in some tissues, aligning with findings that increased HDL is linked to AMD risk [29]	Involved in cholesterol ester transfer, with CETP deficiency leading to high HDL levels [29]	Used in genetic studies to understand lipid metabolism pathways in AMD [29]	The exact mechanisms of how gene expression regulation relates to AMD progression need further elucidation [29]
*C2/CFB*	AMD susceptibility loci involved in the complement system. Variants are significantly related to progression to advanced AMD [31]	C2 and CFB are components of the complement pathway.	Included in predictive models for AMD prevalence and incidence [31]	Genetic risk predictions in multifactorial diseases like AMD have limitations [41]
*CFI*	Susceptibility locus for AMD, part of the complement system. Predicted lower expression of CFI in AMD cases compared to controls [29]	Regulates the alternative complement pathway [29]	Contributes to understanding the genetic basis of AMD [29]	Gene expression in diseased tissue may differ significantly from healthy tissue, which current transcriptome-wide association study (TWAS) models do not fully capture [29]
*RLPB1*	Predicted to have lower gene expression in AMD cases in retinal tissue. One of six genes potentially exclusive to retina affects [29]	Encodes cellular retinaldehyde-binding protein 1; mutations cause diseases such as retinitis punctata albescens and rod-cone dystrophy [29]	Understanding retina-specific effects in AMD etiology [29]	Changes in retinal gene expression can only partly explain genome-wide association study (GWAS) association signals. Data on RPE or choroid tissue gene expression is not yet available to draw further conclusions [29]

### 1.2. Environmental Risk Factors and Pathogenic Mechanisms

In addition to genetic predisposition, several environmental factors such as aging, cigarette smoking, high-fat diet, and chronic light exposure contribute significantly to the onset and progression of dry AMD. Among these, smoking is the strongest modifiable risk factor, promoting oxidative stress, chronic inflammation, and vascular dysfunction within the retina [42]. High-fat diets exacerbate lipid dysregulation and drusen accumulation, while prolonged exposure to blue light induces phototoxic injury to RPE cells via reactive oxygen species (ROS) generation [43,44,45]. Environmental insults also interact with gene expression through epigenetic mechanisms such as DNA methylation, histone modification, and non-coding RNA regulation. Although the epigenetic landscape of AMD remains incompletely defined, accumulating evidence suggests that epigenetic alterations modulate genetic and metabolic pathways, amplifying or modifying disease phenotypes [46]. Together, these environmental and molecular interactions underscore the multifactorial nature of dry AMD pathogenesis.

### 1.3. Pathological Hallmarks of Dry AMD

Early pathological changes in dry AMD arise predominantly in the parafoveal region, where rod photoreceptors are abundant [47]. These include RPE cell enlargement or multinucleation and increased accumulation of bisretinoids [9,48]. As the disease progresses, RPE cells degenerate, detach from BrM, or migrate into the subretinal space [49]. With aging, changes in RPE cause accumulation of basal deposits in BrM containing apolipoprotein B100. When these deposits become thick and contain heterogeneous debris, basal laminar deposits (BLamD) form between the RPE and its basement membrane, while basal linear deposits (BLinD) form within the inner collagenous layer of BrM. Clinically nodular BLinD and focal basal BLamD appear as soft drusen [50]. Concurrent choriocapillaris degeneration compromises oxygen and nutrient supply to the RPE and photoreceptors (Figure 1) [51]. Subretinal drusenoid deposits (SDDs) are also commonly detected and may serve as predictive biomarkers for GA progression [52].

## 2. Events and Cellular Pathways in Dry AMD

The pathogenesis of dry AMD arises from a multifaceted interplay of cellular dysfunctions and molecular signaling pathways. Although genetic predispositions and environmental exposures establish the foundation of disease susceptibility, it is the convergence of oxidative stress, lipid dysregulation, chronic inflammation, and mitochondrial impairment that ultimately drives retinal degeneration and the formation of hallmark features such as drusen and GA. In this section, we highlight the major cellular events that contribute to RPE dysfunction, photoreceptor loss, and choriocapillaris compromise, and we discuss how these processes integrate to shape disease progression and inform potential therapeutic targets.

### 2.1. Oxidative Stress

The high metabolic demands of photoreceptors and RPE cells lead to continuous production of reactive oxygen species (ROS), making oxidative stress one of the central pathogenic mechanisms in dry AMD [53]. Environmental exposures—including cigarette smoke and high-fat diets—further exacerbate oxidative injury within the retina [54]. The macula is especially susceptible due to its high oxygen consumption, abundance of polyunsaturated fatty acids, constant light exposure, and enrichment of endogenous photosensitizers [55,56]. Among these photosensitizers, bisretinoids that accumulate in RPE lysosomes act as potent generators of photooxidative stress. Upon light exposure, bisretinoids form singlet oxygen and superoxide anions (Figure 2), which damage essential biomolecules, including lipids, proteins, and nucleic acids [56]. Lipid peroxidation products, such as malondialdehyde (MDA), form covalent adducts and oxidation-specific epitopes frequently detected in AMD tissues [54]. Chronic oxidative stress contributes directly to early AMD pathology by impairing RPE metabolism, disrupting lysosomal function, and initiating drusen biogenesis.

### 2.2. Lipid Polymorphism and Lipid Dysregulation

Lipid dysregulation is another major contributor to dry AMD development. Genome-wide association studies (GWAS) have paradoxically associated elevated high-density lipoprotein (HDL) cholesterol with increased AMD risk. Drusen—the hallmark extracellular deposits of AMD—contain abundant lipids, including esterified cholesterol, phosphatidylcholine, and oxidized lipoproteins [57]. Genetic polymorphisms in lipid-related genes such as lipase C (*LIPC*), cholesteryl ester transfer protein (*CETP*), ATP-binding cassette transporter A1 (*ABCA1*), and *APOE* contribute to altered lipid transport and impaired clearance in the RPE–BrM interface [50]. Under normal physiology, the RPE mediates cholesterol efflux as HDL-like particles toward the choroid and photoreceptors [58]. When cholesterol trafficking is disrupted, APOB100-containing lipoproteins accumulate within BrM [59], where they undergo oxidation and integrate with hydroxyapatite and inflammatory proteins, forming the lipid–protein-rich cores of drusen [60]. Thus, aberrant lipid accumulation, combined with genetic susceptibility, drives the formation of extracellular deposits central to early dry AMD.

### 2.3. Inflammation and Immune Activation

Chronic inflammation is a defining component of AMD pathogenesis. Drusen frequently harbor complement proteins—including C3, C5, and the membrane attack complex (MAC, C5b-9)—that localize within the subretinal space and choriocapillaris pillars. The common CFH Y402H polymorphism impairs the regulatory function of CFH, promoting sustained complement activation and chronic inflammation [61,62]. During complement dysregulation, the anaphylatoxin C5a can traverse BrM and activate inflammatory and pro-angiogenic pathways [63]. C-reactive protein (CRP), often elevated in AMD, normally partners with CFH to clear cellular debris [64], but the 402H variant exhibits reduced CRP-binding affinity, further exacerbating uncontrolled complement activity [62]. Together, genetic predispositions and persistent immune activation drive sustained complement imbalance, contributing to RPE injury and extracellular deposit accumulation characteristic of dry AMD.

### 2.4. Mitochondrial Dysfunction

Mitochondrial abnormalities are consistently observed in AMD-affected RPE cells. Structural alterations—including disrupted cristae, outer membrane fragmentation, and swollen mitochondria—compromise ATP production, calcium regulation, and redox signaling. Mitochondrial DNA (mtDNA) mutations, particularly those affecting complexes I and III, accumulate preferentially in the macula and often precede photoreceptor degeneration [65].

Defective mitophagy and impaired mitochondrial biogenesis further amplify oxidative stress, resulting in excess ROS production and insufficient recycling of damaged mitochondria [66,67,68]. Antioxidant defense pathways—including mitochondrial heat shock protein 70 (mtHsp70), uncoupling protein 2 (UCP2), and superoxide dismutase 3 (SOD3)—are significantly downregulated in AMD RPE cells [69]. Collectively, these mitochondrial impairments reduce cellular resilience to metabolic and oxidative insults, contributing to progressive RPE degeneration in dry AMD.

### 2.5. Autophagy and Drusen Biogenesis, and RPE–Choriocapillaris Interdependence

Autophagy plays a critical role in RPE homeostasis by facilitating the clearance of metabolic waste, damaged organelles, and photoreceptor-derived debris. Impaired autophagy in the RPE disrupts the removal of photoreceptor outer segment material and dysfunctional mitochondria, thereby promoting intracellular lipofuscin accumulation and extracellular deposit formation, including drusen [70]. Oxidative stress in RPE and BrM triggers autophagic responses; however, chronic oxidative injury can overwhelm this pathway, leading to the accumulation of oxidized proteins and lipids within drusen. Consistent with this, drusen isolated from AMD patients contain markers of autophagy and exosome-related proteins [71,72]. Genetic and experimental studies further support a protective role of autophagy in AMD. RB1CC1/FIP200, component of the autophagy initiation complex, is expressed in the RPE, and its deletion suppresses autophagy and induces AMD-like features, including subretinal drusenoid deposits, accumulation of oxidized and inflammatory proteins, and microglial activation [73]. Similarly, silencing the core autophagy genes *ATG5* and *BECN1* in RPE cells increases ROS, enhances lipofuscin accumulation, impairs mitochondrial function under oxidative stress, and reduces cell viability [74]. Together, these findings suggest that autophagy mechanistically suppresses drusen biogenesis, a central hallmark of dry AMD.

In addition, the RPE and choriocapillaris form a highly interdependent functional unit essential for outer retinal homeostasis [75]. The choriocapillaris supplies oxygen and metabolites to the RPE and photoreceptors while facilitating the removal of metabolic waste [76], whereas the RPE maintains choriocapillaris integrity through the secretion of trophic factors such as vascular endothelial growth factor (VEGF) [77]. Importantly, it remains uncertain whether primary dysfunction originates in the RPE or the choriocapillaris during early AMD, as degeneration of either component can destabilize the other and converge on shared pathogenic pathways.

Neurotrophic factors further modulate this interdependence. Pigment epithelium-derived factor (PEDF) is a key neurotrophic and anti-angiogenic factor that supports photoreceptor survival, regulates lipid metabolism, and modulates inflammatory signaling pathways implicated in AMD progression [78,79,80]. Disruption of PEDF signaling may therefore exacerbate both neuronal vulnerability and RPE dysfunction, contributing to disease initiation and progression [75].

## 3. Mechanistic Mouse Models of Dry AMD

Mice are the most widely used animal model for investigating AMD pathogenesis. Despite lacking a macula, they can be engineered to recapitulate many histological and molecular features of human dry AMD, including drusen-like sub-RPE deposits and bisretinoid (RPE lipofuscin) accumulation [5,16]. Anatomically, mice share key retinal structures with humans—including the RPE, Bruch’s membrane (BrM), and the choriocapillaris—and their genome is highly amenable to targeted genetic manipulation [61]. Although only 22 genes overlap between the 64 mouse-specific and 171 human-specific RPE-enriched genes, comparative transcriptomic analyses reveal strong conservation of fundamental biological functions and signaling pathways [81].

It has been proposed that the central retina of the mouse resembles peripheral macular regions that are vulnerable during early stages of human AMD. The central mouse retina exhibits higher photoreceptor density and a thinner BrM than the peripheral retina, paralleling the structural characteristics of the human macula, albeit with steeper gradients in humans. Moreover, its high photoreceptor density increases the phagocytic load on the RPE, and its rod-to-cone ratio approximates that of the outer macula [82]. These similarities make the central mouse retina a practical surrogate for modeling early macular degeneration.

Although several environmental and genetic factors are known to increase AMD susceptibility, predicting which individuals will develop the disease remains challenging. Thus, animal models provide essential tools for dissecting the complex mechanisms underlying AMD pathology and for evaluating potential therapeutic interventions [83]. Nevertheless, the absence of a true macula in mice limits their direct applicability to human macular degeneration. To address these limitations, complementary models such as induced pluripotent stem cell (iPSC)-derived RPE organoids, nonhuman primate models, and ex vivo human RPE systems are increasingly being developed to more accurately recapitulate human retinal physiology and disease processes.

In this review, we highlight representative dry AMD mouse models categorized according to their predominant pathogenic mechanisms, including oxidative stress, lipid dysregulation, inflammation and immune dysfunction, drusen-like deposit formation, bisretinoid accumulation, and other mechanistic pathways (Figure 3).

### 3.1. Oxidative Stress-Driven Models

Oxidative stress is a well-established driver of RPE dysfunction and drusen formation in dry AMD [84,85,86]. PGC-1α (peroxisome proliferator-activated receptor γ coactivator-1α) serves as a central regulator of mitochondrial biogenesis and lipid metabolism and additionally suppresses inflammation through inhibition of NF-κB signaling [87,88,89] (Appendix A). Mice heterozygous for *Pgc-1α* and fed a high-fat diet develop hallmark AMD-like features, including BrM thickening, APOE/APOJ-positive drusen-like deposits, increased ROS levels, mitochondrial impairment, bisretinoid accumulation, and defective autophagy [90].

Nuclear E2-related factor 2 (NRF2), a key antioxidative transcription factor (Appendix A), plays a critical role in cellular defense against oxidative stress. *Nrf2* knockout mice exhibit subretinal drusen-like deposits as well as progressive RPE and photoreceptor degeneration [91,92,93]. Combined deletion of *Pgc1α* and *Nrf2* further exacerbates oxidative injury, disrupts mitophagy, and accelerates retinal degeneration, underscoring the synergistic contribution of mitochondrial and antioxidative pathways in AMD pathogenesis [94].

Hypoxia-induced stress has been modeled in *P4htm* (prolyl 4 hydroxylase transmembrane) knockout mice. Loss of *P4htm* destabilizes HIF-1α, resulting in RPE and PR degeneration accompanied by drusen-like deposits [95].

Collectively, these oxidative stress-based models highlight the central role of mitochondrial dysfunction, impaired antioxidative defenses, and increased ROS generation in driving early and progressive pathological features of dry AMD (Table 2).

### 3.2. Lipid Dysregulation-Driven Drusen-like Deposit Models

Disruption of lipid metabolism is closely associated with drusen biogenesis and early dry AMD pathology [96]. Peroxisome proliferator-activated receptors (PPARs)—ligand-activated transcription factors—regulate lipid homeostasis, inflammation, extracellular matrix (ECM) remodeling, and angiogenesis, all of which are relevant to AMD pathogenesis (Appendix A) [97]. Mice deficient in *Pparβ/δ* develop prominent subretinal deposits, BrM thickening, and RPE degeneration, accompanied by increased ApoE and bisretinoid accumulation [97]. Similarly, *Apoe^−/−^* mice exhibit cholesterol accumulation and BLinD-like deposits in the retina, supporting the contribution of impaired lipid transport to early deposit formation [98,99].

Transgenic mice overexpressing *ApoB100*, when fed a high-fat diet and exposed to blue-green light, develop BlamD, RPE vacuolization, and significant alterations in BrM morphology [100,101]. *Col18α1^−/−^* mice show abnormal vitamin A metabolism, sub-RPE collagen accumulation, and progressive visual decline [102], while RPE-specific deletion of chloride intracellular channel 4 (*Clic4)* reveals the RPE as a major source of lipid-derived components of drusen-like deposits [103]. Collectively, these models recapitulate key aspects of lipid-driven pathology and highlight the central role of disrupted lipid trafficking and metabolism in dry AMD (Table 2).

### 3.3. Inflammation and Immune Dysregulation-Driven Models

Dysregulation of the complement system and immune pathways plays a pivotal role in dry AMD pathogenesis [104,105,106]. *Cfh* knockout and *CfhY420H* transgenic mice demonstrate hallmark AMD-like features, including drusen-like lesions, C3 deposition, BrM alterations, and increased bisretinoid accumulation [107,108].

The chemokine receptor 1 (CX3CR1), expressed by microglia, regulates microglial migration and homeostasis. Subretinal microglial accumulation is a recognized feature of human AMD, and both *Cx3cr1^−/−^* [109] and *Ccl2^−/−^/Cx3cr1^−/−^* mice develop subretinal microglial aggregates, RPE degeneration, and early drusen-like deposits [110].

*Cd46* knockout mice—lacking a key complement regulatory protein also present in humans—display uncontrolled complement activation, RPE and photoreceptor degeneration, and characteristic drusen-like lesions [111,112,113].

Together, these inflammation- and immune-driven models highlight how complement dysregulation, microglial activation, and impaired debris clearance collectively contribute to RPE injury, extracellular deposit formation, and progressive dry AMD pathology (Table 2).

### 3.4. RPE Lipofuscin Accumulation Models

Lipofuscin accumulation within the RPE is a defining hallmark of dry AMD [16,114]. In this review, the term lipofuscin specifically refers to bisretinoid-derived RPE lipofuscin, rather than lipofuscin species arising from non-retinal cellular aging. Bisretinoids represent a family of fluorescent vitamin A-derived adducts—including N-retinylidene-N-retinylethanolamine (A2E), N-retinylidene-N-retinyl-glyceryl-phospoethanolamine (A2GPE), all-trans-retinal dimer (atRALdi), and related compounds—that accumulate within RPE lysosomes. These bisretinoids are produced in photoreceptor outer segments via non-enzymatic reactions of all-*trans*- and 11-*cis*-retinal with phosphatidylethanolamine, as retinaldehyde intermediates circulate through the visual cycle. These compounds are subsequently transferred to RPE lysosomes through daily phagocytosis of photoreceptor outer segments. Because bisretinoids are intrinsically photoreactive, their lifetime accumulation and light exposure lead to chronic oxidative stress, lysosomal dysfunction, impaired phagocytosis, and progressive RPE and photoreceptor degeneration (Table 3). The mouse models described below represent distinct mechanistic pathways—ranging from visual cycle dysregulation to impaired degradation and metabolic overload—that drive bisretinoid accumulation and RPE damage in dry AMD.

#### 3.4.1. Genetic Models Involving Visual Cycle Enzymes

*Abca4^−/−^* mice, the classical model for Stargardt disease, accumulate high levels of bisretinoids in the RPE due to impaired clearance of retinaldehyde intermediates, leading to light-induced retinal damage [115,116,117,118]. Mice lacking both Rdh8 and Abca4 (*Rdh8^−/−^/Abca4^−/−^*) further exacerbate this phenotype, exhibiting accelerated bisretinoid accumulation and heightened retinal vulnerability [119]. Similarly, *Nrl^−/−^/Abca4^−/−^* mice demonstrate enhanced cone susceptibility, offering valuable insight into cone-specific degeneration relevant to AMD pathology [120]. Collectively, these models illustrate how disruptions in the visual cycle and retinaldehyde detoxification drive bisretinoid buildup and subsequent RPE and photoreceptor degeneration.

#### 3.4.2. Lipid Metabolism and Photoreceptor Vulnerability

Alterations in photoreceptor lipid composition can indirectly promote bisretinoid accumulation and RPE stress. Mutations in the elongation of very-long-chain fatty acid-like 4 (Elovl4) gene—responsible for synthesizing very-long-chain polyunsaturated fatty acids—lead to elevated levels of A2E and progressive central photoreceptor loss [121]. These findings highlight the critical link between lipid metabolic homeostasis, oxidative imbalance, and the predisposition of photoreceptors and RPE cells to lipofuscin-mediated toxicity.

#### 3.4.3. Impaired Phagocytosis and Retinal Debris Accumulation

Mer tyrosine kinase (MERTK) is essential for RPE phagocytosis of photoreceptor outer segments (POS). In *Mertk^−/−^* mice, impaired phagocytic clearance leads to rapid accumulation of undigested POS material and excessive intracellular bisretinoid formation [122]. This model demonstrates how defective phagocytosis directly contributes to lipofuscin overload, lysosomal dysfunction, and subsequent RPE degeneration.

#### 3.4.4. Environmental and Metabolic Risk Factor Models

Environmental and metabolic conditions, particularly diet, profoundly influence retinal lipid homeostasis and bisretinoid generation. Mice maintained on long-term high-fat diets (HFD) exhibit AMD-like pathology characterized by RPE vacuolization, BrM thickening, BLamD-like deposits, disorganization of photoreceptor outer segments, and accumulation of proteins such as CLUSTERIN and TIMP3 [123]. These phenotypes support clinical observations that dietary lipid burden modulates AMD risk.

HFD-induced obesity has also been linked to increased vitamin A transport and elevated phosphatidylethanolamine levels, thereby enhancing bisretinoid formation within the RPE [44]. Together, these models emphasize the metabolic contribution to lipofuscin accumulation and underscore the potential of dietary interventions in mitigating AMD risk, particularly in genetically susceptible populations.

### 3.5. Immunological and Stress-Induced Mouse Models

Dry AMD is driven not only by chronic oxidative injury but also by dysregulated immune signaling and stress-mediated RPE degeneration. Several non-genetic models have been developed to mimic these processes, many of which recapitulate key pathological features of GA and acute RPE loss (Table 4).

#### 3.5.1. Complement and Immune Activation Models

Complement dysregulation is one of the major genetic and mechanistic contributors to dry AMD. Carboxyethyl pyrrole (CEP), a lipid peroxidation product derived from oxidized docosahexaenoic acid (DHA), has been widely used to induce complement activation and generate drusen-like pathology in mice [124]. Immunization with CEP-modified proteins leads to sub-RPE deposits, BrM alterations, and inflammatory responses reminiscent of early AMD.

Similarly, CD46 knockout mice develop spontaneous complement overactivation, resulting in RPE and photoreceptor degeneration, drusen-like lesions, and chronic inflammatory signaling—hallmarks of dry AMD [97,98,99]. Because these phenotypes arise without exogenous immune stimulation, CD46 deficiency provides a valuable model for studying intrinsic complement imbalance and evaluating complement-targeted therapeutic strategies.

#### 3.5.2. Ferroptosis-Associated Models

Ferroptosis, a regulated form of necrosis driven by iron-dependent lipid peroxidation, has emerged as a potential mechanism contributing to RPE degeneration in GA. Glutathione peroxidase 4 (GPX4), a key enzyme that detoxifies lipid hydroperoxides, plays a crucial protective role in the RPE. RPE-specific Gpx4 knockout mice exhibit widespread RPE cell death, disrupted BrM integrity, and photoreceptor degeneration—features highly consistent with late-stage dry AMD and GA [125,126]. These models provide critical insight into the role of ferroptosis and uncontrolled lipid peroxidation in driving irreversible RPE loss.

#### 3.5.3. Chemically Induced Models

Chemical stressors are frequently used to model acute RPE injury and oxidative stress in vivo [127]. Sodium iodate (NaIO_3_), administered intravenously or subretinally, induces dose-dependent RPE and photoreceptor damage characterized by excessive ROS generation, elevated malondialdehyde (MDA), and increased senescence-associated β-galactosidase (SA-β-Gal) activity [128,129]. Owing to its reproducibility and rapid onset of pathology, NaIO_3_ remains one of the most widely used agents for modeling RPE loss.

Polyethylene glycol (PEG)-induced retinal injury provides an additional model of acute RPE stress. PEG administration leads to RPE atrophy, photoreceptor thinning, autophagy dysfunction, and dysregulation of AMD-related genes [129,130]. These chemically induced models are particularly valuable for studying early RPE degeneration and for evaluating potential cytoprotective or antioxidant therapies.

### 3.6. Emerging Alternatives to Traditional Mouse Models

#### 3.6.1. iPSC-Derived RPE Organoids

Induced pluripotent stem cell (iPSC)-derived retinal pigment epithelial organoids (RPEorg) represent valuable in vitro systems for studying AMD pathogenesis, particularly drusen biogenesis [131]. These 3D cultures can be maintained up to 360 days and develop several AMD-like features, including increased autofluorescence, lipid droplet accumulation, calcification, and extracellular aggregates enriched in drusen-associated proteins, such as apolipoprotein E (APOE) and tissue inhibitor of metalloproteinases-3 (TIMP3).

RPE organoids arise spontaneously during the differentiation of iPSC-derived retinal organoids and recapitulate key features of native RPE, including phagocytic capacity, tight junction formation, and robust apicobasal polarity. They display apical microvilli and a collagen-IV-positive basement membrane resembling Bruch’s membrane (BrM) [132].

Organoids derived from patients with mutations such as PRPF31 exhibit genotype-specific abnormalities, including defective RNA splicing, polarity loss, and impaired phagocytosis. Gene-editing approaches can rescue these phenotypes, restoring structural and functional integrity [133]. These models offer powerful platforms for mechanistic studies and for testing gene correction, drug screening, and cell-based therapies.

Moreover, patient-specific iPSCs are appealing because they contain exact mutation of interest along with the patient’s own genetic makeup, which can affect how that mutation contributes to the condition. A study comparing iPS-derived RPE cells from AMD patients and healthy patients showed that AMD cells formed larger amounts of sub-RPE deposits with a lipid- and protein-rich, drusen-like makeup [134]. Another study showed that iPS-RPE cells from AMD patients carrying ARMS2/HTRA1 risk variant showed higher levels of complement and inflammatory proteins than cells from non-AMD donors [135]. These findings highlight the need to address multiple disease pathways simultaneously and incorporate individual genetic risk factors [136]. Moreover, human iPSC-derived retinal organoids can autonomously form 3D retinal cups containing all major retinal cell types, including photoreceptors that achieve advanced maturation with early outer segment disc formation and light sensitivity, offering a powerful system for disease modeling and potential future therapies [137]. iPSC-based therapies are still in the early stages of development, and unlike conventional small or large molecule drugs, they encounter unique challenges related to safety, potency, genetic stability, immune responses, tumor risk, reproducibility, scalability, and successful engraftment [138].

#### 3.6.2. Nonhuman Primate Models

Nonhuman primates (NHPs), particularly *Macaca fascicularis*, represent the most anatomically and physiologically relevant in vivo models for macular research due to their cone-rich macula and distinct foveal pit—features absent in rodents [139,140]. Targeted genetic NHP models have been generated to study inherited macular diseases, including achromatopsia and Best vitelliform macular dystrophy (BVMD) [139,141]. For example, *Macaca fascicularis* carrying the heterozygous mutation develops progressive macular lesions that closely mimic early BVMD pathology [142]. Owing to their structural similarity to the human macula, NHPs serve as essential platforms for evaluating gene therapy, CRISPR-based gene editing, and stem cell-based interventions aimed at photoreceptor rescue and macular restoration [143,144].

Moreover, the NHP model is valuable for studying AMD pathology, testing preventive and therapeutic agents, and screening candidate compounds, particularly for atrophic AMD. A patent describes sodium iodate administered to cynomolgus monkeys developing key features of human AMD, including GA, RPE degeneration, and photoreceptor damage [145]. This sodium iodate-induced model is extensively used for screening potential therapies for dry AMD [146]. Moreover, blue-light-induced NHP model leads to the progressive degeneration of outer retina; this model differs from the human AMD-associated geographic atrophy because it is induced by cellular stress rather than natural disease progression [147]. NHP models play a key role in the development and testing of new treatments for dry AMD, including gene- and cell-based therapies, by supporting both mechanistic research and preclinical assessment [148]. Despite their significant advantages, disease progression in NHP models is slow, making long-term studies time-consuming, resource-intensive, and costly to maintain [5].

#### 3.6.3. Ex Vivo Human RPE Models

Ex vivo human RPE systems provide practical and ethically favorable approaches for studying AMD-related RPE dysfunction. These models are based on culturing immortalized, primary, or iPSC-derived RPE cells on aged human BrM explants [149]. Such systems enable evaluation of fundamental RPE functions—including adhesion, apoptosis, phagocytosis, and gene expression—under physiologically relevant substrate conditions.

Fetal human RPE (fhRPE) cells more closely resemble healthy native RPE compared with ARPE-19 cells, which exhibit features of aged or stressed RPE [150]. Additionally, coculture systems integrating human RPE with neural retina explants allow investigation of RPE–photoreceptor (PR) interactions, offering insights into PR preservation, synaptic maintenance, and degeneration [151].

These ex vivo approaches also support the 3Rs principles (Replacement, Reduction, Refinement) by reducing the reliance on whole-animal models while enabling high-resolution single-cell analyses [152]. The comparative advantages and limitations of mouse models, in vitro models, and non-human primate models are explained in Table 5.

#### 3.6.4. Other Emerging and Alternative Models

Beyond iPSC-derived systems, NHP models, and ex vivo human RPE preparations, additional experimental platforms are being developed to bridge the limitations of traditional rodent models, which lack a macula. These include advanced rodent strains engineered for improved macular approximation, long-term retinal organ cultures, microphysiological systems, and bioengineered retinal tissues designed to replicate human macular architecture and function more faithfully [153,154,155]. Together, these emerging approaches offer new avenues for studying AMD pathogenesis and evaluating therapeutic strategies.

**Table 5 ijms-27-00202-t005:** Comparative advantages and limitations of mouse models, in vitro systems, and nonhuman primates used in AMD research.

Model System	Key Advantage	Major Limitation
Mouse models	High degree of genetic conservation with humans. Well-established tools for genetic manipulation (CRISPR, KO/KI, transgenics) enabling mechanistic studies of complement dysregulation, lipid metabolism, and oxidative stress. Rapid aging, short reproductive cycles, and relatively low maintenance costs enable large-scale and longitudinal studies. Wide availability of standardized strains and experimental reagents [156].	Lack a macula and fovea, limiting direct modeling of macular-specific degeneration and drusen formation. AMD is multifactorial; most mouse models reproduce single pathogenic pathways rather than full disease complexity. Limited genetic diversity reduces generalizability to heterogeneous human populations. Strain-specific variability can influence phenotype and treatment responses [156].
In Vitro Models (iPSC-RPE, 2D/3D Organoids)	Human-derived RPE and retinal cells capture patient-specific genotypes, including AMD risk variants (CFH, ARMS2). Highly controlled environments allow precise manipulation of oxidative stress, complement activity, and lipid metabolism. Suitable for high-throughput drug screening and mechanistic assays. Reduced animal use offers ethical advantages. 3D retinal organoids partially recapitulate retinal layering and early photoreceptor development [157,158,159].	Unable to fully replicate in vivo retinal microenvironment, including vasculature, immune components, and choroidal support. Limited capacity to model aging, a central driver of AMD pathogenesis. Organoid generation is time-consuming, variable, and costly. Conventional 2D cultures may exhibit altered physiology or incomplete RPE maturation. Systemic interactions contributing to AMD cannot be modeled [157,159].
Nonhuman Primates (NHPs)	Possess a true macula and fovea, closely resembling human retinal anatomy. Retinal physiology, photoreceptor architecture, and choroidal vasculature are highly comparable to humans. Can develop age-related drusen and RPE alterations. Essential for late-stage preclinical validation of pharmacokinetics, delivery strategies, safety, and efficacy. Provide the closest translational bridge between rodent studies and human clinical trials [159,160,161].	Limited availability and very high housing and experimental costs. Long lifespan prolongs aging and disease-induction studies. Substantial ethical constraints due to cognitive capacity and similarity to humans. Genetic manipulation is more difficult and less established than in mice. Species- and protocol-dependent variability may affect reproducibility [159,162,163].

## 4. Dry AMD Therapeutics

Dry AMD, which accounts for nearly 90% of all AMD cases, still lacks a broadly effective treatment capable of halting or reversing disease progression [164]. As summarized in Section 2 and Section 3, multiple converging mechanisms—including complement dysregulation, oxidative injury, mitochondrial dysfunction, chronic inflammation, lipid imbalance, and bisretinoid accumulation—drive progressive RPE and photoreceptor degeneration. Accordingly, therapeutic strategies for dry AMD focus on slowing disease progression, stabilizing retinal structure, or targeting specific pathogenic pathways. This section reviews (i) current FDA-approved treatments and (ii) emerging pipeline therapies that target key mechanisms implicated in dry AMD pathogenesis (Table 6).

### 4.1. Current Approaches

#### 4.1.1. AREDS/AREDS2 Supplements

The age-related eye disease study (AREDS) remains the only widely recommended nutritional intervention shown to slow AMD progression. The original AREDS formulation (vitamins C and E, zinc, beta-carotene, and copper) reduced the risk of developing advanced AMD by 25% and decreased vision loss by 19% in patients with intermediate AMD [165].

AREDS2, which replaced beta-carotene with lutein and zeaxanthin due to safety concerns in smokers, demonstrated an 18% reduction in progression to advanced AMD [166]. As highlighted in Section 3.1, oxidative stress plays a central role in promoting RPE damage and drusen formation; therefore, the antioxidant-based AREDS formulations aim to mitigate this mechanism.

A recent NIH study further reported that daily AREDS supplementation significantly delayed GA progression toward the fovea—particularly in cases with extrafoveal lesions—showing up to a 55% reduction in central GA advancement [167]. Although AREDS cannot prevent the onset of AMD or reverse established GA, it remains the most evidence-based strategy for slowing progression in intermediate-stage disease.

#### 4.1.2. Pegcetacoplan (Syfovre)

Pegcetacoplan, a complement C3 inhibitor, is the first FDA-approved treatment for GA secondary to AMD (Figure 4). In pivotal trials (FILLY, OAKS, DERBY), monthly intravitreal administration resulted in up to a 26% reduction in GA lesion growth compared with sham treatment [168,169]. A phase 4 post-marketing study (NCT06161584) is currently underway to evaluate long-term safety, optimal dosing frequency, and real-world treatment outcomes.

#### 4.1.3. Avacincaptad Pegol (Izervay)

Avacincaptad pegol (ACP) is a pegylated RNA aptamer that specifically inhibits complement C5 (Figure 4). In GATHER II trial (NCT04435366), it reduced GA lesion growth by approximately 27% with a favorable safety profile, leading to FDA approval in 2023. However, a slightly higher incidence of choroidal neovascularization (6.7% vs. 4.1% in sham) was observed, necessitating careful monitoring during treatment [170].

### 4.2. Emerging and Investigational Therapies

#### 4.2.1. Complement Pathway Inhibitors

As described in Section 3.3, chronic complement activation is a major pathogenic driver in dry AMD, contributing to sustained inflammation, RPE dysfunction, and lesion progression. Multiple therapeutic strategies have therefore focused on inhibiting specific components of the complement cascade.

Danicopan, an oral factor D inhibitor, blocks the formation of the alternative pathway C3 convertase and demonstrates favorable ocular biodistribution, including accumulation in the RPE and choroid [171,172]. IONIS FB-LRx, an antisense oligonucleotide targeting complement factor B, effectively reduces circulating factor B levels and is currently under evaluation in the phase II GOLDEN study (NCT03815825) [173].

Other agents target additional branches of the complement system. ANX0007, a recombinant monoclonal antibody directed against C1q, inhibits the classical complement pathway and has demonstrated a favorable safety profile in early clinical trials (NCT04656561) [174]. CB2782 and its PEGylated derivative CB2782-PEG are complement C3-inactivating proteases under early-phase development; PEGylation improves pharmacokinetics and enables less frequent intravitreal dosing [175].

Collectively, these complement-directed agents reflect ongoing efforts to restore complement homeostasis and prevent complement-mediated RPE injury in dry AMD.

#### 4.2.2. Inflammation Modulators and Steroids

Beyond complement inhibition, several emerging therapies target broader inflammatory pathways implicated in dry AMD, including cytokine signaling, inflammasome activation, and autoinflammatory stress responses.

Fludrocortisone acetate, a synthetic mineralocorticoid, demonstrated safety in phase Ib clinical study and may stabilize the blood–retinal barrier while reducing retinal edema without elevating intraocular pressure [176]. Xiflam, an orally active small-molecule inhibitor of Connexin-43 hemichannels, suppresses NLRP3 inflammasome activation and reduces autoinflammatory signaling within the RPE. Preclinical studies have shown that Xiflam preserves retinal morphology and reduces infiltration of inflammatory cells [177,178].

These inflammation-targeted therapies complement inhibitors by attenuating downstream inflammatory cascades, potentially offering additive protection against chronic RPE and photoreceptor injury in dry AMD.

#### 4.2.3. Visual Cycle Modulators

As outlined in Section 3.5, dysregulation of the visual cycle enhances the accumulation of toxic bisretinoids within RPE lysosomes. Therapeutic strategies targeting this pathway aim to limit the formation of such lipofuscin fluorophores and thereby mitigate downstream oxidative and phototoxic injury.

ALK-001, a deuterium-stabilized vitamin A analog, decreases dimerization of retinaldehyde and slows A2E formation. ALK-001 has completed phase III evaluation (SAGA, NCT03845582), with trial results forthcoming [179]. Tinlarebant, an oral retinol-binding protein 4 (RBP4) inhibitor currently undergoing phase III testing in the DRAGON trial, has received FDA Breakthrough Therapy designation for Stargardt disease (STGD1) due to its ability to limit retinoid flux to the retina [180]. Similarly, STG-001—another small-molecule RBP4 suppressor—completed a phase II clinical study (NCT04489511) with a favorable safety profile and has been granted orphan drug designation by both the FDA and the European Medicines Agency (EMA) [181].

In contrast, emixustat, an RPE65 inhibitor and the first visual cycle modulator evaluated clinically for dry AMD, did not significantly slow GA progression in the SEATTLE IIb/III trial (NCT01802866) and was associated with adverse effects such as delayed dark adaptation [182,183]. These outcomes highlight the challenges of directly inhibiting visual cycle enzymes, which can interfere with photoreceptor physiology

Consequently, current development efforts increasingly emphasize indirect modulation—such as controlling retinoid transport through RBP4 inhibition or stabilizing vitamin A metabolism—rather than direct blockade of enzymatic steps. These metabolic rebalancing approaches may offer safer and more effective strategies for reducing bisretinoid accumulation in the RPE.

#### 4.2.4. Neuroprotective and Mitochondrial Protective Agents

Photoreceptor and RPE degeneration in dry AMD are closely linked to mitochondrial dysfunction, oxidative injury, and apoptosis, making neuroprotection and mitochondrial preservation important therapeutic objectives. ONL1204, a first-in-class inhibitor of Fas-mediated photoreceptor apoptosis, demonstrated robust photoreceptor preservation in preclinical studies and completed a phase I clinical trial (NCT04744662) [184]. Another strategy employs ciliary neurotrophic factor (CNTF), delivered using an encapsulated cell therapy device (NT-501 implant), which enables long-term intraocular production of CNTF. Phase II studies have shown that CNTF delivery can stabilize retinal structure and partially preserve visual function [185].

Beyond direct neuroprotection, interventions that enhance mitochondrial bioenergetics or mitigate oxidative stress are gaining increasing attention. Candidate compounds targeting mitochondrial fusion–fission balance, antioxidant response pathways, or mitophagy may reduce RPE vulnerability and complement existing therapeutic approaches. Such agents are being explored as potential combination therapies alongside complement inhibitors, aiming to strengthen cellular resilience and slow the trajectory of neurodegeneration. Together, these neuroprotective and mitochondrial-targeted strategies seek to preserve neuronal integrity and delay irreversible vision loss in dry AMD.

Integrated therapeutic strategies combining complement inhibition with neuroprotective interventions are increasingly recognized as a rational approach for the management of dry AMD [186]. Complement inhibitors primarily target chronic inflammation and immune-mediated RPE injury, whereas neuroprotective strategies aim to preserve photoreceptor and RPE viability downstream of inflammatory damage. This mechanistic complementarity directly reflects the multifactorial nature of dry AMD and supports the rationale for combination therapies. In this context, mesenchymal-stem-cell (MSC)-based approaches have shown promise due to their anti-inflammatory, immunomodulatory, and neuroprotective properties, suggesting potential synergy with complement-targeted therapies in preserving retinal structure and function [187].

#### 4.2.5. Photobiomodulation (PBM)

PBM employs visible to near-infrared light (typically 510–800 nm) to enhance mitochondrial function, promote ATP synthesis, and reduce oxidative and inflammatory stress in retinal cells [188]. Multi-wavelength PBM systems—most notably the Valeda^®^ Light Delivery platform—have demonstrated encouraging outcomes across several clinical studies (LIGHTSITE I–III), including improvements in best-corrected visual acuity (BCVA) and reductions in drusen volume [189,190]. The therapeutic effect appears most pronounced in patients with intermediate AMD, although the durability of these responses and the optimal treatment frequency remain under active investigation.

#### 4.2.6. Gene Therapy and Long-Acting Vector-Based Approaches

Gene therapy, particularly vector-based strategies, is emerging as a promising long-acting therapeutic approach for chronic retinal disorders such as dry AMD. A key advantage of this modality is its potential to overcome limitations associated with repeated intravitreal injections, including treatment burden, poor adherence, and cumulative procedural risk [191]. The eye is well-suited for gene therapy due to its relative immune privilege, compartmentalization, and the ability to achieve sustained intraocular expression with low vector doses [192].

Several gene therapy programs aim to achieve durable complement inhibition. One example is CTX001, an adeno-associated virus (AAV)-based therapy designed to express a soluble truncated form of complement receptor 1 (mini-CR1), thereby inhibiting the alternative complement pathway. Preclinical studies have demonstrated favorable safety and biological activity, supporting its further development for geographic atrophy (GA) secondary to AMD [193]. In parallel, vector-mediated expression of complement-inhibiting nanobodies has emerged as an innovative strategy. Nanobodies, derived from camelid single-domain antibodies, offer high specificity, small size, and stability, enabling precise targeting of complement components such as C3 or upstream factors, including C2 [194,195].

Beyond complement modulation, gene therapy approaches targeting neurotrophic support are also under active investigation. Pigment epithelium-derived factor (PEDF), a potent neurotrophic and anti-angiogenic factor, has demonstrated protective effects against ischemia–reperfusion injury and light-induced photoreceptor degeneration in preclinical models when delivered via gene transfer [196]. Ikarovec is developing a bicistronic gene therapy (IKC159V) designed to express soluble CD46 (sCD46), a membrane regulatory protein that attenuates complement activation. Preclinical data suggest that sustained expression of sCD46 may reduce complement-mediated damage while also limiting VEGF-driven vascular leakage and neovascular responses, highlighting its potential as a long-acting therapeutic candidate for GA [197].

Collectively, these vector-based strategies underscore the growing emphasis on durable, single-administration therapies that address both inflammatory and neurodegenerative components of dry AMD, offering a promising path forward for long-term disease management.

#### 4.2.7. Other Experimental Candidates

A number of emerging small molecules and biologics target alternative mechanisms associated with retinal degeneration beyond complement inhibition. Remofuscin, an orphan-designated small molecule that enhances lysosomal degradation of lipofuscin, demonstrated favorable safety in the phase II Stargardt remofuscin treatment trial (STARTT), suggesting potential applicability to RPE lipofuscin-associated pathology in dry AMD [198].

RO7303359, an anti-IL-33 biologic developed by Roche, completed phase I trial in 2023 (NCT04615325), confirming its tolerability and aiming to modulate inflammation-driven retinal degeneration. Risuteganib, a multifunctional anti-integrin peptide that influences mitochondrial homeostasis and visual transduction pathways, improved visual function in a phase IIb trial, with a phase III trial currently in preparation [199].

Collectively, these investigational agents broaden the therapeutic landscape for dry AMD by targeting mitochondrial dysfunction, lysosomal clearance pathways, and inflammatory signaling. While promising, their clinical development remains challenging due to modest efficacy signals and the inherent heterogeneity of AMD—a topic that will be further addressed in the following section.

### 4.3. Lessons from Clinical Failures

Despite a rapidly expanding therapeutic pipeline, numerous investigational agents have failed to demonstrate meaningful efficacy in slowing or reversing dry AMD progression across clinical stages [200]. Several factors consistently underlie these failures, including suboptimal trial design, inappropriate endpoint selection, and an incomplete understanding of disease heterogeneity [201]. Historically, visual acuity (VA) served as the primary endpoint in ophthalmic clinical trials; however, VA decline typically occurs only in advanced disease and poorly correlates with early pathological changes such as drusen accumulation, RPE dysfunction, or photoreceptor loss [201]. In response, the U.S. FDA has recognized the reduction in geographic atrophy (GA) area as an acceptable surrogate primary endpoint for late-stage dry AMD trials [202]. Nevertheless, even with this more relevant anatomical metric, many trials have not achieved their efficacy goals.

For instance, eculizumab (a monoclonal antibody targeting C5a and C5b; NCT00935883) demonstrated good tolerability but failed to reduce GA lesion enlargement [203]. Lampalizumab, a humanized antibody targeting complement factor D, initially showed encouraging results in phase II studies (NCT01229215) but did not replicate these outcomes in phase III trials (NCT02247479, NCT02247531), leading to program discontinuation [204,205]. One explanation for this limited efficacy is lampalizumab’s selective inhibition of the alternative complement pathway, with minimal influence on the classical and lectin pathways that also contribute to AMD pathology. Lampalizumab selectively targets the alternative complement pathway, leaving classical and lectin pathways largely unaffected, which may explain its limited efficacy despite strong genetic and preclinical rationale. Direct inhibition of RPE65 by emixustat disrupted physiological retinoid cycling, leading to delayed dark adaptation and poor tolerability, which ultimately outweighed its theoretical benefit in reducing bisretinoid formation.

Fenretinide, designed to lower circulating retinol and retinol-binding protein (RBP) and thereby reduce bisretinoid formation, was well tolerated in a phase II trial (NCT00429936), but induced delayed dark adaptation and failed to demonstrate functional or structural benefit [206,207]. Similarly, tanospirone (NCT00890097), a serotonin receptor agonist previously used as an antidepressant, did not reduce GA lesion area in a phase III multicenter study.

The redundancy of the complement system presents a major challenge for therapeutic targeting. Because multiple component pathways are interconnected and can compensate for one another, inhibition of single component often fails to fully suppress downstream inflammation. This likely explains the limited clinical efficacy seen with selective complement inhibitors and points to the need for broader combination-based strategies.

Moreover, therapeutic success in dry AMD is strongly dependent on disease stage. Interventions aimed at modifying disease mechanisms are most likely to be effective in early and intermediate stages, when RPE dysfunction, drusen accumulation, and chronic inflammation are still present but not yet irreversible. In advance stage, characterized by GA and extensive cell loss, suppressing upstream pathways alone is unlikely to restore function, shifting the focus toward neuroprotective or regenerative approaches. Finally, the gap between preclinical models and human disease continues to limit translation. Most animal models capture only isolated aspects of dry AMD and do not fully reflect its slow, multifactorial progression in humans. These differences can lead to overestimation of therapeutic benefit and help explain repeated clinical trial failures.

Collectively, these clinical failures underscore the need for biomarker-driven and mechanism-informed trial designs. Future studies should integrate multimodal endpoints—including retinal structure (OCT), fundus autofluorescence, photoreceptor integrity, and functional measures—to better capture disease activity and therapeutic impact. Aligning clinical endpoints with mechanistic insights from preclinical models will be essential for improving translational success and identifying therapies capable of producing meaningful benefits in patients with dry AMD.

## 5. Conclusions

Age-related macular degeneration is a progressive neurodegenerative disorder of the retina and remains a leading cause of central vision loss in the elderly. Dry AMD, which represents nearly 90% of all AMD cases, is marked by early drusen and RPE lipofuscin (bisretinoid) accumulation, ultimately progressing to geographic atrophy and irreversible photoreceptor degeneration. Despite its prevalence, no curative therapy exists.

Multiple genetic and molecular pathways—including oxidative stress, complement dysregulation, aberrant lipid handling, and mitochondrial dysfunction—collectively drive disease onset and progression [208]. Although the approval of pegcetacoplan represents a significant milestone for complement-targeted therapy, currently available treatments primarily slow rather than halt degeneration. Mechanistic mouse models have played a central role in advancing our understanding of these pathways, successfully recapitulating key pathological features even in the absence of a macula.

Future progress will depend on refining translational models, establishing robust and early-stage biomarkers, and developing combination therapies tailored to both disease stage and genetic background. With continued integration of mechanistic insights and therapeutic innovation, the development of truly disease-modifying treatments for dry AMD is becoming an increasingly achievable goal.

## 6. Future Direction

Advancing dry AMD research will require coordinated progress across disease modeling, therapeutic innovation, and multimodal diagnostic technologies. Integrating these approaches is essential for bridging the gap between mechanistic discovery and clinical translation.

### 6.1. Retinal Organoids in Drug Screening

Human retinal organoids (ROs), derived from hiPSCs or hESCs, provide physiologically relevant 3D systems that recapitulate retinal cell organization, polarity, and layered structure [209]. Their scalability and human origin make them valuable platforms for drug screening, toxicology testing, and mechanistic studies [210]. Recent work has demonstrated that compounds such as metformin and TN1 protect against sodium iodate-induced retinal injury by activating the HMOX1 pathway [211].

More advanced retina-on-a-chip (RoC) platforms integrate multiple retinal cell types with microfluidic perfusion, offering the ability to evaluate drug-induced retinopathies from agents such as chloroquine and gentamicin in a controlled, reproducible environment [209,212]. Together, these organoid- and chip-based systems provide rapid, ethically sound, and translationally relevant tools for target identification and preclinical therapeutic screening.

### 6.2. Emerging Therapies and Clinical Innovation

Innovative therapeutic strategies increasingly target diverse molecular pathways and incorporate neuroprotection, regenerative medicine, and personalized treatment paradigms [213]. Stem cell-based and tissue-engineering approaches aim to replace damaged RPE or photoreceptors with functional cells, showing promise in early-phase clinical programs [214,215]. Gene-editing tools and optogenetic therapies—including ongoing programs at UPMC—seek to restore visual signaling in residual retinal circuitry, potentially benefiting patients with advanced disease [215].

Optogenetic therapy offers an innovative strategy for vision restoration by introducing light-sensitive opsins into surviving inner retinal neurons, particularly bipolar cells (BCs) and retinal ganglion cells (RGCs). Through gene delivery, these genetically modified cells can be activated by defined wavelengths of light—often augmented by light-amplifying devices—thereby bypassing degenerated photoreceptors and enabling downstream transmission of visual signals to the brain. This approach allows precise stimulation of selected retinal neuron populations, supports long-term transgene expression from a single intervention, and exploits largely intact downstream neural circuits. Increasing preclinical and early clinical evidence suggests that optogenetic interventions may provide a viable therapeutic option for advanced retinal degenerative conditions, including late-stage disease, where photoreceptors are lost, but inner retinal architecture remains preserved [216]. Notably, in 2021, the first clinical evidence demonstrated that a patient with retinitis pigmentosa regained object recognition following optogenetic therapy using AAV2-ChrimsonR in combination with training via light-enhancing goggles [217]. Despite this promise, several technical challenges remain, including optimization of opsin sensitivity, efficient and cell-type-specific gene delivery, and integration with external optical control systems. Continued advances in opsin engineering, vector design, and bioelectronic interfaces will be essential for the successful clinical translation of optogenetic therapies [218].

The successful translation of personalized therapies for dry AMD is constrained by regulatory hurdles and the complexity of effective patient stratification. Personalized medicine aims to tailor therapeutic strategies to individual patient profiles; however, its implementation is challenged by evolving regulatory requirements and difficulties in defining appropriate patient subgroups [219]. The regulatory framework governing personalized medicine remains multifaceted and is still maturing. In the United States, the FDA oversees personalized therapies through CDER (Center for Drug Evaluation and Research), CDRH (Center for Devices and Radiological Health), and CBER (Center for Biologics Evaluation and Research). However, existing statutory frameworks do not fully accommodate the interdependent nature of drugs, diagnostics, and biologics, leading to regulatory inconsistencies. To address these challenges, the FDA is developing clearer processes and center-specific policies to improve oversight and streamline premarket review of diagnostics, including molecular assays and imaging-based tools. Ongoing efforts also focus on advancing ethical and methodological standards in clinical pharmacogenomics and strengthening inter-center coordination to support more efficient joint reviews and balanced regulation across platforms [220]. In AMD, effective patient stratification is critical because disease progression and therapeutic response vary widely across disease stages, including GA. This heterogeneity complicates efficacy assessment and the design of adequately powered clinical trials. Emerging AI-driven analyses of OCT imaging hold promise for identifying disease-specific biomarkers and functional endpoints, thereby supporting more precise patient stratification and improved trial design in AMD [221].

Parallel advances in computational tools are transforming clinical research: artificial intelligence models enable automated imaging analysis, prediction of disease trajectories, and improved patient stratification for trials [213]. Projects such as MACUSTAR are establishing standardized functional and structural endpoints for intermediate AMD, helping optimize future clinical trial design and outcome measures [222].

### 6.3. Longitudinal Multimodal Imaging

Multimodal imaging (MMI) has become central to precise disease staging and monitoring of AMD progression [223]. The integration of OCT, OCT angiography, fundus autofluorescence, and color fundus photography provides complementary information on drusen morphology, RPE atrophy, photoreceptor integrity, and GA lesion dynamics [224,225,226]. Longitudinal MMI enables patient-specific progression tracking and can validate objective structural and functional endpoints for clinical fundus autofluorescence (FA) studies [227,228].

In addition to imaging- and AI-based approaches, omics-derived biomarkers and functional endpoints provide critical information for AMD research by enabling patient stratification and the identification of trial-specific disease signatures [229]. Omics approaches interrogate the molecular landscape of AMD across genetic, transcriptomic, proteomic, and metabolomic layers. Genomic studies account for a substantial proportion of AMD heritability, whereas findings from non-genomic omics studies often require further validation due to limited sample sizes and cohort heterogeneity. Functional endpoints, including electroretinography, dark adaptation testing, and microperimetry, offer complementary measures of retinal performance that may detect disease-related changes not fully captured by structural imaging alone. Integration of multi-omics data with functional assessments and imaging biomarkers holds promise for improving predictive accuracy, refining clinical trial endpoints, and enabling personalized therapeutic strategies in dry AMD [230].

The integration of high-resolution imaging, AI-based analytics, omics biomarkers and next-generation disease models—including retinal organoids and NHP macular systems—will accelerate the development of precise, mechanism-guided therapies. Ultimately, this ecosystem of technologies is poised to advance the discovery of effective, disease-modifying treatments for dry AMD.

## Figures and Tables

**Figure 1 ijms-27-00202-f001:**
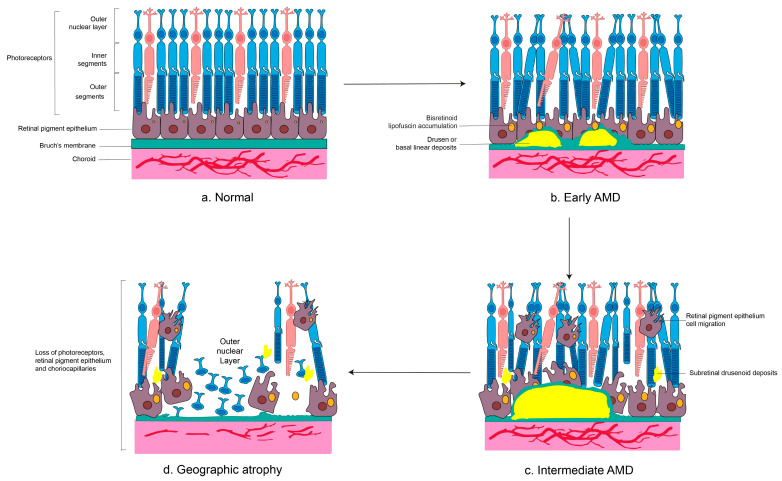
Pathological hallmarks of dry age-related macular degeneration (AMD). Dry AMD progresses from early drusen formation and intracellular lipofuscin accumulation to RPE degeneration, photoreceptor loss, and choriocapillaris thinning, ultimately leading to geographic atrophy.

**Figure 2 ijms-27-00202-f002:**
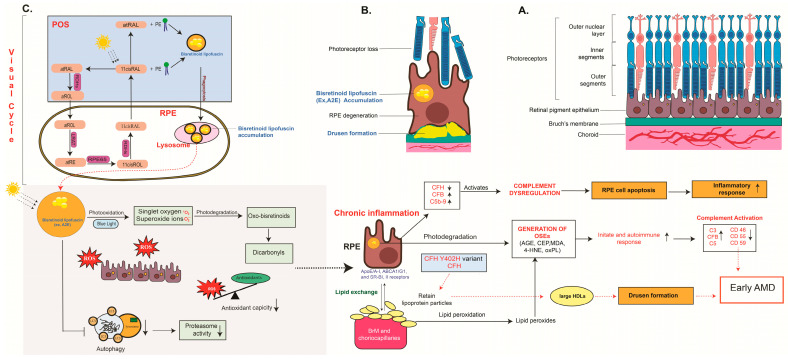
Pathomechanistic processes in dry AMD. Bisretinoid lipofuscin accumulation, phototoxic stress, lipid dysregulation, and complement activation converge to induce oxidative injury, lysosomal dysfunction, and chronic inflammation in the RPE. A. Normal retina structure showing photoreceptors, retinal pigment epithelium (RPE), Bruch’s membrane (BrM), and choroid. B. In early dry AMD, bisretinoid lipofuscin (e.g., A2E) accumulates in RPE lysosomes, with concomitant drusen formation and photoreceptor stress. C. Bisretinoids such as A2E, generated via the visual cycle, accumulate in RPE lysosomes. Upon light exposure, A2E undergoes photooxidation and photodegradation, producing singlet oxygen and superoxide radicals, leading to the generation of reactive degradation products. These photooxidative processes further impair lysosomal function and proteasome activity, reduce antioxidant capacity, and initiate chronic inflammation in RPE cells. This cascade contributes to two major pathogenic axes: (1) Complement dysregulation, initiated by downregulation of CFH and upregulation of CFB and C5b-9, promoting RPE cell apoptosis and inflammatory amplification. (2) Generation of oxidation-specific epitopes (OSEs), including advanced glycation end products (AGEs), carboxyethylpyrrole (CEP), malondialdehyde (MDA), 4-hydroxynonenal (4-HNE), and oxidized phospholipids (oxPLs)—arising from both A2E photodegradation and lipid peroxidation. These OSEs stimulate innate and autoimmune responses, driving further complement activation (e.g., C3, CFB, C5 upregulation) and suppression of complement inhibitors (CD46, CD55, CD59). In parallel, RPE-derived lipoprotein particles accumulate in BrM due to impaired efflux (e.g., via ApoE/A1, ABCA1, SR-BI) and compromised CFH regulation, especially in the Y402H risk variant. This leads to the formation of large HDL-like deposits (HLDs) with proinflammatory properties that further exacerbate early AMD lesion formation, such as drusen. RPE, retinal pigment epithelium; POS, photoreceptor outer segment; A2E, N-retinylidene-N-retinylethanolamine; *at*RAL, all-trans-retinal; *at*ROL, all-trans-retinol; *at*RE, all-*trans*-retinyl ester; 11*cis*ROL, 11-*cis*-retinol; 11*cis*RAL, 11-*cis*-retinal; BrM, Bruch’s membrane; ROS, reactive oxygen species; RDHs, retinol dehydrogenase; LRAT, lecithin retinol acyl transferase; RPE65, retinal pigment epithelium-specific 65 kDa protein.

**Figure 3 ijms-27-00202-f003:**
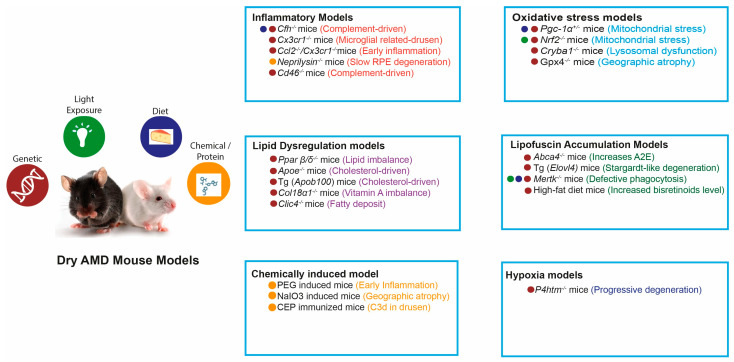
Overview of mouse models recapitulating features of dry AMD. Representative models are categorized by their major mechanisms, including oxidative stress, lipid dysregulation, complement activation, and bisretinoid accumulation.

**Figure 4 ijms-27-00202-f004:**
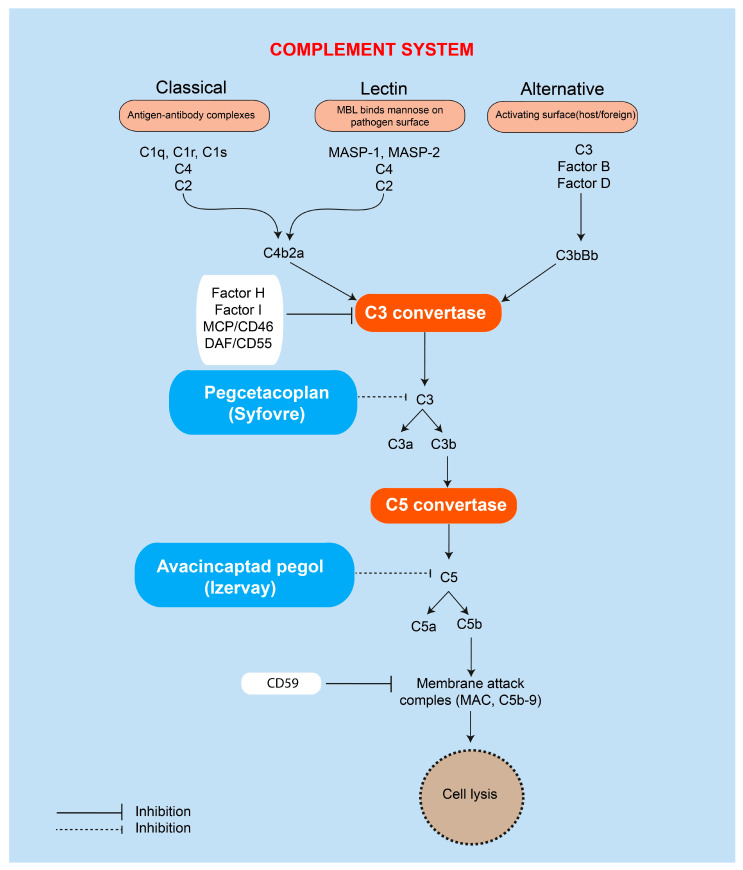
Complement pathways and therapeutic targets in geographic atrophy (GA). FDA-approved inhibitors (pegcetacoplan and avacincaptad pegol) act at C3 and C5 to suppress complement-mediated retinal damage.

**Table 2 ijms-27-00202-t002:** Summary of Mouse Models for Drusen-like Deposits in Dry Age-related Macular Degeneration (AMD).

Mouse Model	Key Pathological Feature	Impact Summary
**Oxidative Stress**		
*Pgc-1α^+/−^*	Drusen, lipofuscin, mitochondrial dysfunction, RPE migration	Mitochondrial stress model
*Nrf2^−/−^*	ROS ↑, drusen, RPE degeneration	Oxidative stress-driven AMD
*P4htm^−/−^*	Drusen, photoreceptor shortening, RPE abnormalities	Age-progressive retinal degeneration
**Lipid Dysregulation**		
*Pparβ/δ^−/−^*	Subretinal deposits, RPE degeneration, Apoe upregulation, ERG changes.	Lipid imbalance and RPE loss
*Apoe^−/−^*	Hypercholesterolemia, Bruch’s membrane alterations	Cholesterol-associated AMD
Tg (Apob100)	BrM thickening, BLamD, RPE vacuolization	Diet and light-sensitive model
*Col18α1^−/−^*	BLamD-like deposits, impaired vitamin A metabolism	Vitamin A dysregulation model
*Clic4^−/−^*	Fat accumulation, RPE abnormalities	Fatty deposit model
**Inflammation and Immunity**		
*Cfh^−/−^*	Complement activation, subretinal deposits, vision loss	Complement-driven AMD
*Cx3cr1^−/−^*	Subretinal microglia, drusen-like lesions	Microglia-associated drusen
*Ccl2^−/−^/Cx3cr1^−/−^*	Drusen, inflammation, photoreceptor atrophy	Early-onset inflammatory AMD
*Cd46^−/−^*	Drusen formation, PR loss, autophagy dysregulation	Complement and autophagy dysregulation

RPE indicates retinal pigment epithelium; ROS, reactive oxygen species; Apoe, apolipoprotein E; ERG, electroretinography; BrM, Bruch’s membrane; BLamD, basal laminar deposits; ↑, increases.

**Table 3 ijms-27-00202-t003:** Experimental Mouse Models for Lipofuscin Accumulation in Dry Age-related Macular Degeneration.

Mouse Model	Key Pathological Feature	Impact Summary
**Lipofuscin Accumulation**		
*Abca4^−/−^*	Lipofuscin accumulation (↑ A2E); mild rod degeneration	Stargardt disease type1 (STGD1) Model
Tg (Elovl4)	Lipofuscin accumulation (↑ A2E); undigested phagosomes; RPE atrophy; photoreceptor loss	Stargardt-like degeneration (STGD3) model
*Mertk^−/−^*	Lipofuscin accumulation (↑ A2E); impaired phagocytosis; photoreceptor degeneration	Phagocytosis defect model
High-fat diet	Lipofuscin accumulation (↑ A2E); RPE vacuolization; disorganized PROS; BlamD-like deposits	Diet-induced AMD features

RPE indicates retinal pigment epithelium; BLamD, basal laminar deposits; PROS, photoreceptor outer segment; ↑, increases.

**Table 4 ijms-27-00202-t004:** Alternative Models of Dry Age-related Macular Degeneration: Non-genetic, Chemical, and Immune-mediated Approaches.

Mouse Model	Inductive Process	Key Pathological Feature	Impact Summary
**Oxidative Stress**			
NaIO_3_-induced	Chemical	Macrophage infiltration, PR apoptosis, ↑ ROS/MDA, ONL thinning	Oxidative damage model
*Gpx4^−/−^*	Genetic	Progressive RPE loss, ↓ visual function	Geographic atrophy model
**Inflammation**			
PEG-induced	Chemical	ONL thinning, RPE hypopigmentation, ↑ autophagy (ATG12), drusen-like deposits	Acute inflammatory model
CEP-immunized	Immunological	Anti-CEP antibodies, C3d in BrM, sub-RPE drusen, RPE lesions	Immune-mediated drusen model

ROS indicates reactive oxygen species; MDA, malondialdehyde; ONL, outer nuclear layer; RPE, retinal pigment epithelium; BrM indicates Bruch’s membrane; ↑ increases; ↓ decreases.

**Table 6 ijms-27-00202-t006:** Current and Investigational Therapies for Dry Age-related Macular Degeneration.

Drug Name	Target	Status	Phase
**Nutraceuticals**			
AREDS/AREDS2	Antioxidants	As a dietary supplement	
**Complement Inhibitors**			
Pegcetacoplan	C3	FDA approval in April 2023, Recruiting	Phase IV
Avacincaptad pegol (ACP)	C5	Completed FDA approval for the treatment of GA.	Phase III
Danicopan	Factor D	Ongoing	Phase II
IONIS-FB-LRx	Factor B	Active, not recruiting	Phase II
ANX0007	C1Q	Active, recruiting	Phase II
CB2782-PEG	C3	Development Phase	Preclinical study
CB2782	C3	Development Phase	Preclinical study
**Inflammation Modulators**			
Fludrocortisone acetate	PLA_2_	Completed	Phase Ib
Xiflam	NLRP3 inflammasome	No any update	Expected phase II
**Visual Cycle Modulator** **s**			
ALK-001	Vit-A dimerization	Active, not recruiting	Phase III
Tinlarebant	RBP4	Ongoing	Phase III
STG-001	RBP4	Completed	Phase II
Emuixustat	RPE65	Completed	Phase III
**Neuroprotective Agents**			
ONL 1204	CD95	Completed	Phase I
CNTF	Neuroprotection (PRs and RPE cells)	Completed	Phase II
**Photobiomodulation**			
Valeda^®^ Light Delivery System	Cytochrome C oxidase	Active clinical research	Phase II/III
**Others**			
Remofusin	Lipofuscin granules	Completed	Phase II
RO7303359	IL-33	Completed	Phase I
Risuteganib	Integrin heterodimers	Completed	Phase II

GA indicates geographical atrophy; PLA_2_, phospholipase A2; RBP4, retinol-binding protein 4; RPE65, retinal pigment epithelium-specific 65; IL-33, interleukin 33.

## Data Availability

No new data were created or analyzed in this study. Data sharing is not applicable to this article.

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
