# Peer review of "Integrative Landscape of Dry AMD Pathogenesis, Models, and Emerging Therapeutic Strategies"

_ijms, 2025, doi:10.3390/ijms27010202_

Round 1
Reviewer 1 Report
Comments and Suggestions for Authors
The review summarizes current knowledge on AMD pathogenesis, mechanistic models, and therapeutic insights. It is a valuable resource for researchers/clinicians, but it would benefit from a deeper analysis and a more integrated translational perspective.
- This review is mainly descriptive and fails to include a detailed account of the limitations of current models and therapies. For example, why do complement inhibitors show modest efficacy? The authors should discuss why lampalizumab (a complement factor D antibody) and emixustat (an RPE65 inhibitor) failed despite a strong preclinical rationale.
- The authors overemphasized the importance of mouse models in AMD research, despite their lack of a macula, and have highlighted this limitation. Consider expanding the roles of iPSC-derived RPE organoids (lines 657-669) and non-human primates (lines 418-428). Adding a table that highlights the pros/cons of mouse models versus in vitro models/NHPs can be helpful for readers.
- Expand on synergistic strategies, such as complement inhibition and neuroprotection.
- There is one line on optogenetic therapies (line 675) – this is a promising approach and must be expanded in this review.
- The authors mention imaging and AI (lines 684-694), but not omics-based biomarkers or functional endpoints.
Minor comments:
- Lots of older studies cited – incorporate recent omics and single-cell transcriptomics studies in the references.
- Clarify the difference between lipofuscin and bisretinoids.
- Expand the discussion on regulatory challenges and patient stratification for personalized therapy.
Author Response
Reviewer1
Major comments
The review summarizes current knowledge on AMD pathogenesis, mechanistic models, and therapeutic insights. It is a valuable resource for researchers/clinicians, but it would benefit from a deeper analysis and a more integrated translational perspective.
Response: We thank the reviewer for the positive evaluation of our review and for emphasizing the need for deeper analysis and a more integrated translational perspective. In response to these comments, we have extensively revised the manuscript to move beyond a descriptive overview and to provide a more critical, mechanistically informed, and translationally oriented synthesis of dry AMD research.
Specifically, we have strengthened the critical analysis of current experimental models and therapeutic strategies by explicitly addressing their limitations, including the modest efficacy and clinical failures of several complement- and visual cycle–targeted therapies. We have rebalanced the discussion of disease models by expanding human-relevant systems such as iPSC-derived RPE organoids and non-human primate models, and by introducing a comparative table that clarifies the strengths and limitations of each model in a translational context.
In addition, we have broadened the therapeutic landscape to include synergistic and long-acting strategies, such as combination approaches integrating complement inhibition with neuroprotection, gene-based therapies, and optogenetic interventions. We further enhanced the translational perspective by incorporating discussions on omics-based biomarkers, functional endpoints, patient stratification, and regulatory challenges relevant to personalized therapy development.
Collectively, these revisions provide a more integrated framework that links molecular mechanisms, experimental models, and therapeutic development, thereby strengthening the translational relevance of the review for both researchers and clinicians in the field of dry AMD.
- This review is mainly descriptive and fails to include a detailed account of the limitations of current models and therapies. For example, why do complement inhibitors show modest efficacy? The authors should discuss why lampalizumab (a complement factor D antibody) and emixustat (an RPE65 inhibitor) failed despite a strong preclinical rationale.
Response: We thank the reviewer for this important and insightful comment. We agree that the initial version of the manuscript was primarily descriptive and did not sufficiently emphasize the limitations of translational challenges of current models and therapeutic strategies.
In response, we have substantially revised the manuscript to incorporate a more critical analysis of the mechanistic and translational factors underlying the modest efficacy of complement inhibitors and the clinical failures of lampalizumab and emixustat.
Specifically, we have expanded Section 4.3 (Lessons from Clinical Failures) to discuss:
(i) redundancy and pathway compensation within the complement system,
(ii) stage-dependent therapeutic windows in dry AMD, and
(iii) discrepancies between preclinical models and human disease progression.
For lampalizumab, we now highlight that selective inhibition of complement factor D targets only the alternative pathway, while the classical and lectin pathways remain active, which may limit clinical efficacy despite genetic and preclinical rationale.
For emixustat, we elaborate that direct inhibition of RPE65 disrupts physiological visual cycle function, resulting in adverse effects such as delayed dark adaptation and poor tolerability, ultimately outweighing its theoretical benefit in reducing bisretinoid formation.
These additions strengthen the critical perspective of the review and provide important lessons for future therapeutic development in dry AMD.
Revision: Section 4.3 has been expanded to include a critical discussion of the mechanistic and translational limitations underlying clinical failures (lines 746-760).
“Lampalizumab selectively targets the alternative complement pathway, leaving classical and lectin pathways largely unaffected, which may explain its limited efficacy despite strong genetic and preclinical rationale.” (lines 734-737).
“Direct inhibition of RPE65 by emixustat disrupted physiological retinoid cycling, leading to delayed dark adaptation and poor tolerability, which ultimately outweighed its theoretical benefit in reducing bisretinoid formation.” (lines 737-739)
- The authors overemphasized the importance of mouse models in AMD research, despite their lack of a macula, and have highlighted this limitation. Consider expanding the roles of iPSC-derived RPE organoids (lines 657-669) and non-human primates (lines 418-428). Adding a table that highlights the pros/cons of mouse models versus in vitro models/NHPs can be helpful for readers.
Response: We thank the reviewer for this constructive and insightful comment. We agree that although mouse models have been invaluable for mechanistic studies, their lack of a macula represents a fundamental limitation for modeling human AMD.
In response, we have revised the manuscript to rebalance the discussion of disease models by expanding the sections on iPSC-derived RPE organoids and non-human primate (NHP) models, emphasizing their complementary roles in addressing macula-specific structure, function, and disease progression. We have also refined the language in the mouse model sections to more explicitly acknowledge their translational limitations.
In addition, we have added a new comparative table summarizing the strengths and limitations of mouse models, iPSC-derived in vitro systems, and NHP models, with respect to anatomical relevance, mechanistic insight, scalability, and translational applicability. We believe these revisions provide a more balanced and reader-oriented framework and strengthen the translational perspective of the review.
Revision: The sections on iPSC-derived RPE organoids (Section 3.6.1, lines 459-474) and non-human primate (NHP) models (Section 3.6.2., lines 486-498) have been expanded to emphasize their macula-relevant anatomical and translational advantages.
In addition, a new comparative table has been added to summarize the respective strengths and limitations of mouse models, iPSC-derived in vitro systems, and NHP models in dry AMD research.
Table 5. Comparative advantages and limitations of mouse models, in vitro systems, and non-human primates used in AMD research.
|
Model System |
Key Advantages |
Major Limitations |
|
Mouse models |
High degree of genetic conservation with humans. Well-established tools for genetic manipulation (CRISPR, KO/KI, transgenics) enabling mechanistic studies of complement dysregulation, lipid metabolism, and oxidative stress. Rapid aging, short reproductive cycles, and relatively low maintenance costs enable large-scale and longitudinal studies. Wide availability of standardized strains and experimental reagents[1]. |
Lack a macula and fovea, limiting direct modeling of macular-specific degeneration and drusen formation. AMD is multifactorial; most mouse models reproduce single pathogenic pathways rather than full disease complexity. Limited genetic diversity reduces generalizability to heterogeneous human populations. Strain-specific variability can influence phenotype and treatment responses [1].
|
|
In Vitro Models (iPSC-RPE, 2D/3D Organoids) |
Human-derived RPE and retinal cells capture patient-specific genotypes, including AMD risk variants (CFH, ARMS2). Highly controlled environments allow precise manipulation of oxidative stress, complement activity, and lipid metabolism. Suitable for high-throughput drug screening and mechanistic assays. Reduced animal use offers ethical advantages. 3D retinal organoids partially recapitulate retinal layering and early photoreceptor development [2-4].
|
Unable to fully replicate in vivo retinal microenvironment, including vasculature, immune components, and choroidal support. Limited capacity to model aging, a central driver of AMD pathogenesis. Organoid generation is time-consuming, variable, and costly. Conventional 2D cultures may exhibit altered physiology or incomplete RPE maturation. Systemic interactions contributing to AMD cannot be modeled [2,4].
|
|
Non-Human Primates (NHPs) |
It possesses a true macula and fovea, closely resembling human retinal anatomy. Retinal physiology, photoreceptor architecture, and choroidal vasculature are highly comparable to humans. Can develop age-related drusen and RPE alterations. Essential for late-stage preclinical validation of pharmacokinetics, delivery strategies, safety, and efficacy. Provide the closest translational bridge between rodent studies and human clinical trials [4-6]. |
Limited availability and very high housing and experimental costs. Long lifespan prolongs aging and disease-induction studies. Substantial ethical constraints due to cognitive capacity and similarity to humans. Genetic manipulation is more difficult and less established than in mice. Species- and protocol-dependent variability may affect reproducibility [4,7,8].
|
- Expand on synergistic strategies, such as complement inhibition and neuroprotection.
Response: Thank you for your valuable suggestion. We agree that synergistic strategies are particularly relevant for dry AMD given its multifactorial pathophysiology. In response, we have expanded the manuscript to include a more detailed discussion on integrated therapeutic approaches that combine complement inhibition with neuroprotective strategies, highlighting their mechanistic complementarity and translational potential.
Specifically, we discuss how complement inhibition may attenuate chronic inflammatory injury, while concurrent neuroprotective interventions aim to preserve RPE and photoreceptor integrity, thereby addressing both upstream inflammatory drivers and downstream neuronal loss. These additions emphasize the rationale for combination therapies as a future direction in dry AMD treatment.
Revision: We have added new content in lines 644-653.
Integrated therapeutic strategies combining complement inhibition with neuroprotective interventions are increasingly recognized as a rational approach for the management of dry AMD [9]. Complement inhibitors primarily target chronic inflammation and immune-mediated RPE injury, whereas neuroprotective strategies aim to preserve photoreceptor and RPE viability downstream of inflammatory damage. This mechanistic complementarity directly reflects the multifactorial nature of dry AMD and supports the rationale for combination therapies. In this context, mesenchymal-stem-cell (MSC)-based approaches have shown promise due to their anti-inflammatory, immunomodulatory, and neuroprotective properties, suggesting potential synergy with complement-targeted therapies in preserving retinal structure and function [10].
- There is one line on optogenetic therapies (line 675) – this is a promising approach and must be expanded in this review.
Response: We thank the reviewer for highlighting the importance of optogenetic therapies. We agree that this is a promising and rapidly advancing approach for vision restoration. In response, we have expanded the section 6.2 (Emerging Therapies and Clinical Innovation) to provide a more comprehensive discussion of optogenetic strategies, including their mechanistic rationale, current clinical evidence, and remaining translational challenges.
Revision: New content has been added in lines 812-830.
Optogenetic therapy offers an innovative strategy for vision restoration by introducing light-sensitive opsins into surviving inner retinal neurons, particularly bipolar cells (BCs) and retinal ganglion cells (RGCs). Through gene delivery, these genetically modified cells can be activated by defined wavelengths of light-often augmented by light-amplifying devices—thereby bypassing degenerated photoreceptors and enabling downstream transmission of visual signals to the brain. This approach allows precise stimulation of selected retinal neuron populations, supports long-term transgene expression from a single intervention, and exploits largely intact downstream neural circuits. Increasing preclinical and early clinical evidence suggests that optogenetic interventions may provide a viable therapeutic option for advanced retinal degenerative conditions, including late-stage disease, where photoreceptors are lost but inner retinal architecture remains preserved [11]. Notably, in 2021, the first clinical evidence demonstrated that a patient with retinitis pigmentosa regained object recognition following optogenetic therapy using AAV2-ChrimsonR in combination with training via light-enhancing goggles [12]. Despite this promise, several technical challenges remain, including optimization of opsin sensitivity, efficient and cell-type–specific gene delivery, and integration with external optical control systems. Continued advances in opsin engineering, vector design, and bioelectronic interfaces will be essential for the successful clinical translation of optogenetic therapies [13].
- The authors mention imaging and AI (lines 684-694), but not omics-based biomarkers or functional endpoints.
Response: We thank the reviewer for this insightful comment. We agree that, in addition to imaging and artificial intelligence–based approaches, omics-derived biomarkers and functional endpoints are increasingly important for disease stratification and outcome assessment in dry AMD. In response, we have expanded the relevant section to include a discussion of genomic and multi-omics biomarkers, as well as functional endpoints, highlighting how these measures complement imaging and AI-based analyses in patient stratification, biomarker discovery, and clinical trial design.
Revision: New content has been added in lines 865-877.
In addition to imaging- and AI-based approaches, omics-derived biomarkers and functional endpoints provide critical information for AMD research by enabling patient stratification and the identification of trial-specific disease signatures [14]. Omics approaches interrogate the molecular landscape of AMD across genetic, transcriptomic, proteomic, and metablomic layers. Genomic studies account for a substantial proportion of AMD heritability, whereas findings from non-genomic omics studies often require further validation due to limited sample sizes and cohort heterogeneity. Functional endpoints, including electroretinography, dark adaptation testing, and microperimetry, offer complementary measures of retinal performance that may detect disease-related changes not fully captured by structural imaging alone. Integration of multi-omics data with functional assessments and imaging biomarkers holds promise for improving predictive accuracy, refining clinical trial endpoints, and enabling personalized therapeutic strategies in dry AMD [15].
Minor comments
- Lots of older studies cited – incorporate recent omics and single-cell transcriptomics studies in the references.
Response: We thank the reviewer for this constructive comment. We agree that recent omics and single-cell transcriptomics studies provide important insights into AMD pathogenesis. Accordingly, we have incorporated recent bulk transcriptomic, single-cell RNA-sequencing, and multi-omics studies into the future direction sections to better reflect current advances in the field.
Revision: Recent omics and single-cell transcriptomics studies have been added and cite omics- and single-cell transcriptomics studies have been incorporated and cited (lines 865-877)
- Clarify the difference between lipofuscin and bisretinoids
Response: We thank the reviewer for this comment. We agree that lipofuscin and bisretinoids are not synonymous and that clarification is important for conceptual precision. In the revised manuscript, we now explicitly state that lipofuscin refers to a heterogeneous autofluorescent intracellular material, whereas bisretinoids are chemically defined fluorophores generated from visual cycle–derived retinaldehyde condensation reactions.
We further clarify that the scope of this review is specifically focused on RPE lipofuscin derived from bisretinoids, which are the primary contributors to fundus autofluorescence observed clinically, rather than on lipofuscin as a broad heterogeneous entity.
Revision: Clarifying text has been added to the Introduction (lines 50-56)
- Expand the discussion on regulatory challenges and patient stratification for personalized therapy.
Response: We thank the reviewer for this important suggestion. We have expanded the manuscript to provide a more detailed discussion of regulatory challenges and patient stratification in the context of personalized therapies for dry AMD. The revised section addresses issues related to trial design, endpoint validation, biomarker qualification, and regulatory acceptance of stratified therapeutic approaches.
Revision: Expanded discussion has been added (lines 831-851).
The successful translation of personalized therapies for dry AMD is constrained by regulatory hurdles and the complexicity of effective patient stratification. Personalized medicine aims to tailor therapeutic strategies to individual patient profiles; however, its implementation is challenged by evolving regulatory requirements and difficulties in defining appropriate patient subgroups [16]. The regulatory framework governing personalized medicine remains multifaceted and is still maturing. In the United States, the FDA oversees personalized therapies through CDER (Center for Drug Evaluation and Research), CDRH (Center for Devices and Radiological Health), and CBER (Center for Biologics Evaluation and Research). However, existing statutory frameworks do not fully accommodate the interdependent nature of drugs, diagnostics, and biologics, leading to regulatory inconsistencies. To address these challenges, the FDA is developing clearer processes and center-specific policies to improve oversight and streamline premarket review of diagnostics, including molecular assays and imaging-based tools. Ongoing efforts also focus on advancing ethical and methodological standards in clinical pharmagenomics and strengthening inter-center coordination to support more efficient joint reviews and balanced regulation across platforms [17]. In AMD, effective patient stratification is critical because disease progression and therapeutic response vary widely across disease stages, including GA. This heterogeneity complicates efficacy assessment and the design of adequately powered clinical trials. Emerging AI-driven analyses of OCT imaging hold promise for identifying disease-specific biomarkers and functional endpoints, thereby supporting more precise patient stratification and improved trial design in AMD [18].

Reviewer 2 Report
Comments and Suggestions for Authors
The review is in large well-written covering many pertinent aspects of the complex field of dry AMD biology. The overview of animal models and therapeutic are usefull to the reader.
I have som minor comments:
29-30: While true in some regards, AMD is not typically denoted a neurodegenerative disorder.
89-91: the wording is a bit confusing reagering the causative factor of basal deposit formation
107: "all AMD in the dry form" - what do you mean exactly?
Section 1.1 and 1.4 is similar in some regards and could optimally be combined.
Section 2. Events and cellular pathways ... : The involved pathways are complex. One could add a discussion of autophagy and its role in drusen biogenesis, the relative importance of choriocapillaris dysfunction (it is proabably uncertain if RPE dysfunction or choriocapillaris are primary events), and the role of neurotrophic factors such as PEDF that has been recently reviewed.
Section on therapeutic landscape: What about gene therapies, e.g. CTX0011? Long-acting therapies will probably be paramount for this chronic disease. Various complement inhibitors can be expressed using vector-delivery. Vector-expression of complement-inhibiting nanobodies has also recently been published. PEDF is also a promising neurotrophic factor being developed for gene therapeutic amplication by ikarovec and a lot of preclinical work has been done.
589: "smodulate"
Author Response
Major comments
The review is in large well-written covering many pertinent aspects of the complex field of dry AMD biology. The overview of animal models and therapeutic are usefull to the reader.
Minor comments
Please, find attached the document with some minor revisions.
- 29-30: While true in some regards, AMD is not typically denoted a neurodegenerative disorder.
Response: We thank the reviewer for this comment. We agree that, although neurodegenerative features may be observed in advanced stages, AMD is not typically classified as a primary neurodegenerative disorder. Accordingly, we have revised the text to remove the term “progressive neurodegenerative” and to describe AMD more accurately as a complex, multifactorial retinal disease.
Revision: The term “progressive neurodegenerative” has been removed from the sentence “Age-related macular degeneration (AMD) is a multifactorial and progressive neurodegenerative disorder of the central retina and a leading cause of irreversible vision loss among the elderly”to more accurately reflect the current clinical and pathological classification of AMD (lines 29-30).
- 89-91: the wording is a bit confusing reagering the causative factor of basal deposit formation
Response: We thank the reviewer for this valuable comment regarding the clarity of the causal relationship underlying basal deposit formation. We agree that the original wording may have been ambiguous. In response, we have revised the text to clarify the mechanistic sequence leading to basal laminar deposits (BLamD) and basal linear deposits (BLinD), avoiding an oversimplified causal interpretation.
Revision: The sentence implying a direct causal relationship between RPE degeneration and basal deposit formation has been revised to improve clarity and reflect the multifactorial nature of BLamD and BLinD development (lines 130-135).
- 107: "all AMD in the dry form" - what do you mean exactly?
Response: We thank the reviewer for this comment and apologize for the ambiguity in the original wording. By this statement, we intended to indicate that early and intermediate stages of AMD are classified as the non-neovascular (dry) form, which may subsequently progress to advanced stages. We have revised the sentence to clarify this distinction and avoid potential misunderstanding.
Revision: For clarification, the sentence has been revised and replaced by “The non-neovascular (dry) form of AMD typically encompasses the early and intermediate stages of the disease and may progress through these stages prior to the development of advanced AMD.” (Lines 83-85)
- Section 1.1 and 1.4 is similar in some regards and could optimally be combined.
Response: We thank the reviewer for this helpful comment. We agree that Sections 1.1 and 1.4 contain overlapping content. To improve clarity and avoid redundancy, we have merged the relevant material from Section 1.4 into Section 1.1 and revised the structure accordingly.
Revision: Overlapping content in Sections 1.1 and 1.4 has been consolidated by merging Section 1.4 into Section 1.1, resulting in a more streamlined presentation (lines 76–110).
- Section 2. Events and cellular pathways ... : The involved pathways are complex. One could add a discussion of autophagy and its role in drusen biogenesis, the relative importance of choriocapillaris dysfunction (it is proabably uncertain if RPE dysfunction or choriocapillaris are primary events), and the role of neurotrophic factors such as PEDF that has been recently reviewed.
Response: Thank you for this insightful comment. We agree that the cellular pathways involved in dry AMD are highly complex and that additional discussion of autophagy, choriocapillaris dysfunction, and neurotrophic factors would strengthen the manuscript.
In response, we have added a new subsection (Section 2.5) discussing the role of impaired autophagy in drusen biogenesis, highlighting its contribution to defective clearance of cellular debris and oxidative stress in the RPE. We also revised the text to emphasize the current uncertainty regarding whether RPE dysfunction or choriocapillaris degeneration represents the primary initiating event, underscoring their reciprocal and interdependent relationship.
In addition, we expanded the discussion to include the role of neurotrophic factors, particularly pigment epithelium–derived factor (PEDF), in maintaining retinal homeostasis and modulating inflammatory and degenerative pathways relevant to AMD progression.
Revision: A new section (Section 2.5) has been added in lines 214-245.
2.5 Autophagy and drusen biogenesis, and RPE–choriocapillaris interdependence
Autophagy plays a critical role in retinal pigment epithelium (RPE) homeostasis by facilitating the clearance of metabolic waste, damaged organelles, and photoreceptor-derived debris. Impaired autophagy in the RPE disrupts the removal of photoreceptor outer segment material and dysfunctional mitochondria, thereby promoting intracellular lipofuscin accumulation and extracellular deposit formation, including drusen [1]. Oxidative stress in RPE and BrM triggers autophagic responses; however, chronic oxidative injury can overwhelm this pathway, leading to the accumulation of oxidized proteins and lipids within drusen. Consistent with this, drusen isolated from AMD patients contain markers of autophagy and exosome-related proteins [2,3]. Genetic and experimental studies further support a protective role of autophagy in AMD. RB1CC1/FIP200, component of the autophagy initiation complex, is expressed in the RPE, and its deletion suppresses autophagy and induces AMD-like features, including subretinal drusenoid deposits, accumulation of oxidized and inflammatory proteins, and microglial activation [4]. Similarly, silencing the core autophagy genes ATG5 and BECN1 in RPE cells increases ROS, enhances lipofuscin accumulation, impairs mitochondrial function under oxidative stress, and reduces cell viability [5]. Together, these findings suggest that autophagy mechanistically suppresses drusen biogenesis, a central hallmark of dry AMD.
In addition, the RPE and choriocapillaris form a highly interdependent functional unit essential for outer retinal homeostasis [6]. The choriocapillaris supplies oxygen and metabolites to the RPE and photoreceptors while facilitating the removal of metabolic waste [7], whereas the RPE maintains choriocapillaris integrity through the secretion of trophic factors such as vascular endothelial growth factor (VEGF) [8]. Importantly, it remains uncertain whether primary dysfunction originates in the RPE or the choriocapillaris during early AMD, as degeneration of either component can destabilize the other and converge on shared pathogenic pathways.
Neurotrophic factors further modulate this interdependence. Pigment epithelium–derived factor (PEDF) is a key neurotrophic and anti-angiogenic factor that supports photoreceptor survival, regulates lipid metabolism, and modulates inflammatory signaling pathways implicated in AMD progression [9-11]. Disruption of PEDF signaling may therefore exacerbate both neuronal vulnerability and RPE dysfunction, contributing to disease initiation and progression. [6].
- Section on therapeutic landscape: What about gene therapies, e.g. CTX0011? Long-acting therapies will probably be paramount for this chronic disease. Various complement inhibitors can be expressed using vector-delivery. Vector-expression of complement-inhibiting nanobodies has also recently been published. PEDF is also a promising neurotrophic factor being developed for gene therapeutic amplication by ikarovec and a lot of preclinical work has been done.
Response: We thank the reviewer for this important and forward-looking comment. We agree that long-acting therapeutic strategies, particularly gene-based approaches, are likely to be critical for the management of dry AMD as a chronic disease.
In response, we have expanded the therapeutic landscape section by adding a dedicated subsection on gene therapy and vector-based long-acting approaches. This section now discusses AAV-mediated complement inhibition (including CTX-based strategies), vector-delivered complement-inhibiting nanobodies, and gene therapeutic amplification of neurotrophic factors such as PEDF.
These additions highlight how sustained intraocular expression of therapeutic proteins may overcome the limitations of repeated intravitreal injections and improve long-term disease control in dry AMD.
Revision: We have included the following section (Section 4.2.7) in lines 663-694.
4.2.7 Gene therapy and long-acting vector-based approaches
Gene therapy, particularly vector-based strategies, is emerging as a promising long-acting therapeutic approach for chronic retinal disorders such as dry AMD. A key advantage of this modality is its potential to overcome limitations associated with repeated intravitreal injections, including treatment burden, poor adherence, and cumulative procedural risk [12]. The eye is well suited for gene therapy due to its relative immune privilege, compartmentalization, and the ability to achieve sustained intraocular expression with low vector doses [13].
Several gene therapy programs aim to achieve durable complement inhibition. One example is CTX001, an adeno-associated virus (AAV)–based therapy designed to express a soluble truncated form of complement receptor 1 (mini-CR1), thereby inhibiting the alternative complement pathway. Preclinical studies have demonstrated favorable safety and biological activity, supporting its further development for geographic atrophy (GA) secondary to AMD [14]. In parallel, vector-mediated expression of complement-inhibiting nanobodies has emerged as an innovative strategy. Nanobodies, derived from camelid single-domain antibodies, offer high specificity, small size, and stability, enabling precise targeting of complement components such as C3, or upstream factors including C2 [15,16].
Beyond complement modulation, gene therapy approaches targeting neurotrophic support are also under active investigation. Pigment epithelium–derived factor (PEDF), a potent neurotrophic and anti-angiogenic factor, has demonstrated protective effects against ischemia–reperfusion injury and light-induced photoreceptor degeneration in preclinical models when delivered via gene transfer [17]. Ikarovec is developing a bicistronic gene therapy (IKC159V) designed to expresses soluble CD46 (sCD46), a membrane regulatory protein that attenuates complement activation. Preclinical data suggest that sustained expression of sCD46 may reduce complement-mediated damage while also limiting VEGF-driven vascular leakage and neovascular responses, highlighting its potential as a long-acting therapeutic candidate for GA [18].
Collectively, these vector-based strategies underscore the growing emphasis on durable, single-administration therapies that address both inflammatory and neurodegenerative components of dry AMD, offering a promising path forward for long-term disease management.
- 589: "smodulate"
Response: Thank you for pointing this out. We have corrected the typographical error.
Revision: The typographical error “smodulate” has been corrected to “modulate” (line 703).

Round 2
Reviewer 1 Report
Comments and Suggestions for Authors
The authors have done a commendable job answering all the questions